

Atmospheric
Measurement
Techniques



# Observation of cirrus clouds with GLORIA during the WISE campaign: detection methods and cirrus characterization

**Irene Bartolome Garcia**[1], **Reinhold Spang**[1], **Jörn Ungermann**[1], **Sabine Griessbach**[2], **Martina Krämer**[1], **Michael Höpfner**[3], and **Martin Riese**[1]

[1]Institut für Energie und Klimaforschung (IEK-7), Forschungszentrum Jülich GmbH, 52428 Jülich, Germany
[2]Jülich Supercomputing Centre (JSC), Forschungszentrum Jülich GmbH, 52428 Jülich, Germany
[3]Institute of Meteorology and Climate Research, Karlsruhe Institute of Technology, 76021 Karlsruhe, Germany

**Correspondence:** Irene Bartolome Garcia (i.bartolome@fz-juelich.de)

**Abstract.** Cirrus clouds contribute to the general radiation budget of the Earth and play an important role in climate projections. Of special interest are optically thin cirrus clouds close to the tropopause due to the fact that their impact is not yet well understood. Measuring these clouds is challenging as both high spatial resolution as well as a very high detection sensitivity are needed. These criteria are fulfilled by the infrared limb sounder GLORIA (Gimballed Limb Observer for Radiance Imaging of the Atmosphere). This study presents a characterization of observed cirrus clouds using the data obtained by GLORIA aboard the German research aircraft HALO during the WISE (Wave-driven ISentropic Exchange) campaign in September and October 2017. We developed an optimized cloud detection method based on the cloud index and the extinction coefficient retrieved at the microwindow 832.4–834.4 $cm^{-1}$. We derived macro-physical characteristics of the detected cirrus clouds such as cloud top height, cloud bottom height, vertical extent and cloud top position with respect to the tropopause. The fraction of cirrus clouds detected above the tropopause is on the order of 13 % to 27 %. In general, good agreement with the clouds predicted by the ERA5 reanalysis dataset is obtained. However, cloud occurrence is $\approx$ 50 % higher in the observations for the region close to and above the tropopause. Cloud bottom heights are also detected above the tropopause. However, considering the uncertainties, we cannot confirm the formation of unattached cirrus layers above the tropopause.

## 1 Introduction

High clouds composed of ice crystals are formed in the upper troposphere, where the temperatures are lower than −30 °C. According to Sassen et al. (2008), these high clouds cover 16.7 % of the Earth's surface on average. All these clouds (from now on simply cirrus) are important, due to their frequent occurrence and their effect on the radiative budget of the Earth (Liou, 1986). Cirrus clouds generally have a strong infrared greenhouse effect, leading to warming. However, optically thick cirrus can have the opposite effect (Krämer et al., 2016). Thus, they influence the amount of solar radiative energy received and also the loss of energy. These clouds are challenging to measure because they can appear in multilayered cloud systems and they can be optically very thin, which complicates their detection by active as well as passive nadir sounders. Whereas in situ measurements are capable of detecting the thinnest clouds, they only capture a temporally and spatially limited snapshot. Because of these difficulties, and despite being the subject of many studies, processes related to cirrus clouds are still not well understood and cause large uncertainties in climate projections (IPCC, 2013). Important factors influencing these uncertainties are ice water content, crystal number concentration and size distribution (Fusina et al., 2007). Other important factors that are problematic to determine are exact altitude and thickness. According to Sassen and Cho (1992), cirrus clouds are defined as (optically) thick for an optical depth $\tau > 0.3$, (optically) thin for $0.03 < \tau < 0.3$ and subvisible (SVC) for $\tau < 0.03$ in the visible wavelength region.

Of special interest is the effect of cirrus clouds in the upper troposphere–lowermost stratosphere (UTLS) region. Even small changes in the concentration of water vapor in this region affect the radiative forcing of the atmosphere (Riese et al., 2012). The presence of cirrus clouds above the tropopause, which will evaporate as soon as they experience a temperature increase and thus contribute to the water vapor budget, is still an ongoing discussion. While Dessler (2009) indicated the existence of a substantial amount of cirrus clouds above the tropopause, Pan and Munchak (2011), using the same Cloud-Aerosol Lidar with Orthogonal Polarization (CALIOP) data, demonstrated that the amount of cirrus clouds above the tropopause strongly depends on the definition of the tropopause. Moreover Pan and Munchak (2011) concluded that there is not enough evidence of clouds above the tropopause in the mid-latitudes. A follow up study by Spang et al. (2015), using the measurements from the Cryogenic Infrared Spectrometers and Telescopes (CRISTA) and ERA-Interim temperature fields for the determination of the local tropopause, concluded that there is a significant number of occurrences in the lowermost stratosphere at mid- and high latitudes. A recent study with the Michelson Interferometer for Passive Atmospheric Sounding (MIPAS) and Cloud-Aerosol Lidar and Infrared Pathfinder Satellite Observations (CALIPSO) by Zou et al. (2020) found that CALIPSO observed occurrence frequencies of about 2 % of stratospheric cirrus clouds at mid- and high latitudes and 4 %–5 % for MIPAS at mid-latitudes (6 year mean global distribution 2006–2012). Other studies based on measurements by ground-based lidars show thin cirrus that are unambiguously located in the lowermost stratosphere (Keckhut et al., 2005). In the analysis of Goldfarb et al. (2001) using data from northern mid-latitudes, cirrus cloud tops often occur at the tropopause and SVC constitute 23 % of the total occurrences of cirrus clouds.

Detection of optically thin cirrus clouds and SVCs is a challenge due to the high vertical resolution needed and high sensitivity. Martins et al. (2011) analyzed CALIOP measurements over 2.5 years and gave insight into the global occurrence of SVCs, which are more common in the tropics (30 %–40 %). Reverdy et al. (2012) reported a significant population of SVCs in the tropical upper troposphere. However, Davis et al. (2010) found that CALIPSO would be missing about 2/3 of SVCs with $\tau < 0.01$. Due to the long path of the line of sight (LOS) through the cirrus, typical of limb instruments, clouds that might be invisible to the nadir viewing instruments are detectable. Spang et al. (2008) detected optically thin clouds with ice water content (IWC) down to 0.01 ppmv using the airborne limb instrument Cryogenic Infrared Spectrometers and Telescopes for the Atmosphere – New Frontiers (CRISTA–NF). This IWC matches the lower limit of the expected IWC for mid-latitude cirrus clouds, 0.01–200 ppmv (Luebke et al., 2016). Our study uses data from the airborne Gimballed Limb Observer for Radiance Imaging of the Atmosphere (GLORIA) instrument (Riese et al., 2014; Friedl-Vallon et al., 2014). This instrument possesses the technical characteristics necessary for the detection of thin cirrus and SVCs. It has a spatial sampling of 140 m × 140 m (horizontal sampling × vertical sampling) at a tangent point altitude (i.e., closest point of the LOS to the Earth's surface) of 10 km for a flight altitude of 15 km. It measures in the infrared spectral region between 780 and 1400 cm$^{-1}$ and its long LOS provides sufficient sensitivity to low ice concentrations. The mid-IR radiative properties of cirrus clouds depend on particle size, particle shape and the considered wavelength (Baran, 2005; van de Hulst, 1958; Yang et al., 2001). For this study, the configuration of the retrieval of the extinction coefficient is fixed for the microwindow 832.4–834.4 cm$^{-1}$.

Our work analyzes the cirrus measured by GLORIA during the Wave-driven ISentropic Exchange (WISE) campaign in September and October 2017 with the purpose of obtaining more information about the nature of cirrus (both macro-physical and micro-physical properties) and thus improve the understanding of their formation processes. The analysis includes the macro-physical properties of cirrus clouds, i.e., cloud top height (CTH), cloud bottom height (CBH), vertical extent and their position with respect to the tropopause. The tropopause was computed following the definition of the first thermal tropopause from WMO (1957).

## 2 Datasets and instrument

### 2.1 The instrument: GLORIA

GLORIA is part of the heritage of CRISTA–NF, which was a limb viewing airborne instrument with a vertical resolution of 200–400 m and two spectrometers with spectral resolution of $\approx 2$ and $\approx 1$ cm$^{-1}$, respectively. This instrument represented an important stepping stone toward future remote sensing limb instruments with even higher vertical and horizontal resolution. The GLORIA instrument and the data processing chain is described in previous studies, therefore the reader is referred to the works of Kleinert et al. (2014), Friedl-Vallon et al. (2014), Riese et al. (2014) and Ungermann et al. (2015) for a more detailed description. Here, the main concepts are presented.

GLORIA is an infrared limb emission sounder that combines Fourier-transform spectroscopy with a 2D infrared detector and measures radiances in the mid-IR range (780–1400 cm$^{-1}$). It was designed with the purpose of providing information about trace gases and temperature fluctuations in the observational gap that comprises small-scale structures of less than 500 m of vertical extent and less than 100 km in the horizontal. With GLORIA, it is possible to retrieve the distribution of different trace gases and aerosols, reconstruct gravity waves and study clouds in the UTLS (e.g., Blank, 2013; Krisch et al., 2018; Höpfner et al., 2019). The high spatial resolution, 140 × 140 m (horizontal sampling × vertical

sampling) at a tangent point altitude of 10 km and observer altitude of 15 km, and the high precision sensors to obtain a good pointing accuracy make GLORIA a perfect instrument for investigating optically and vertically thin cirrus. The instrument is typically configured to use $48 \times 128$ pixels of its 2D detector array. Since the main focus of this study is the characterization of cirrus clouds close to the tropopause and thus the most important feature is the vertical resolution, we do not analyze each individual pixel, but the horizontally averaged spectrum (averaged over 48 pixels) of each line of the 2D array. The final result is one profile for each measured set of interferograms with 128 spectra. The amount of radiance that each pixel receives is determined by the point spread function (PSF). The shape of the PSF is approximated by an Airy disk with an aperture of the instrument of 3.6 cm and using $830\,cm^{-1}$ as a reference wavelength. This configuration has been computed from a theoretical setup of the instrument and was validated by cloud top measurements. The vertical sampling is higher the closer the tangent point altitude is to the observer altitude, as the projection of the PSF gets wider the further away the tangent point is. For example, if the observer altitude is 14.7 km at a tangent point of 13 km, the vertical sampling is about 88 m; at 10 km, it is about 150 m and at 8 km it is about 179 m.

GLORIA always points towards the horizon from the right side of the plane. It is typically configured to one of three measuring modes: one high spectral resolution mode called chemistry mode (CM) and two modes, premier and panorama modes (DM), focusing on dynamical effects in the atmosphere. During CM, the instrument is fixed at 90° with respect to the flight trajectory. During the premier and panorama mode, the instrument changes its viewing direction between 45 and 135° in steps of 4 and 2°, respectively, which gives the possibility of observing the same volume of air from different perspectives and thus allowing for tomographic studies. This capability of GLORIA was used for the reconstruction of gravity waves (Krisch et al., 2018) and clouds (Ungermann et al., 2020). Table 1 summarizes the most important technical features of GLORIA. The data processing chain of GLORIA consists of three stages: the raw data processing (level 0), the processing into geolocated calibrated spectra (level 1) and the retrieval of geophysical quantities leveraging the fast radiative transfer model JURASSIC2 (Sect. 2.4) (Hoffmann et al., 2008; Griessbach et al., 2013; Ungermann et al., 2015). This work uses level 1 and level 2 products.

## 2.2 The campaign: WISE

The data analyzed in this study were measured during the WISE (Wave-driven ISentropic Exchange) campaign. It took place in Shannon, Ireland (52.70° N, 8.86° W), in September and October of 2017. With a total of 15 scientific flights (plus a first test flight) (Fig. 1) covering the North Atlantic area, it aims to answer questions related to mixing, the role

**Table 1.** Instrument specifications (Friedl-Vallon et al., 2014). Observer altitude of 15 km and tangent altitude of 10 km.

| Property | Value |
|---|---|
| Temporal sampling | 2 s ($\approx 0.5$ km)/12.8 s ($\approx 3.2$ km) for DM/CM |
| Spectral coverage | 780–1400 $cm^{-1}$ |
| Spectral sampling | 0.0625 $cm^{-1}$ to 0.625 $cm^{-1}$ |
| Detector array size | $256 \times 256$ pixels |
| Used detector array size | $48 \times 128$ pixels |
| Vertical sampling | 0.031°, equal to 140 m |
| Horizontal sampling | 0.031°, equal to 140 m |
| Vertical spatial coverage | $-3.3$° below horizon to 0.8° above horizon |
| Horizontal spatial coverage | 1.5° ($= 48 \times 0.031$°), equal to 6.7 km |
| Yaw pointing range | 45 to 135° |
| Pointing precision (vertical) | 0.012°, equal to $\approx 50$ m (1$\sigma$) |
| Pointing accuracy* | 0.1° |

* Ungermann (2021).

of Rossby wave breaking events in the transport of trace gases, such as water vapor, the formation of cirrus clouds and several other topics (Riese et al., 2017). All the measurements were taken onboard the German research aircraft HALO (High Altitude and Long Range Research Aircraft), where GLORIA was placed in the belly pod. HALO can fly to a maximum altitude of 15 km, which means that the vertical coverage of GLORIA observations during this campaign ranged from $\sim 15$ down to $\sim 5$ km.

## 2.3 Meteorological dataset

We used the high-resolution ERA5 dataset provided by the European Centre for Medium-Range Weather Forecasts (ECMWF). The reanalysis data are available at 31 km horizontal resolution at 137 levels from surface to 80 km (Hersbach et al., 2020). The ERA5 dataset provides hourly data for a large variety of meteorological and climate variables. To perform the comparison between model and measurements, the variables of interest were sampled according to the GLORIA measuring geometry, as shown in Fig. 2. This figure represents the limb geometry during one measurement. For every LOS, every 30 km, the meteorological variables were computed from the corresponding parameters of the ERA5 dataset, i.e., the first thermal tropopause (TP), equivalent latitude and ice water content (IWC), i.e, the cloud ice mass in unit volume of atmospheric air. As the signal is inte-

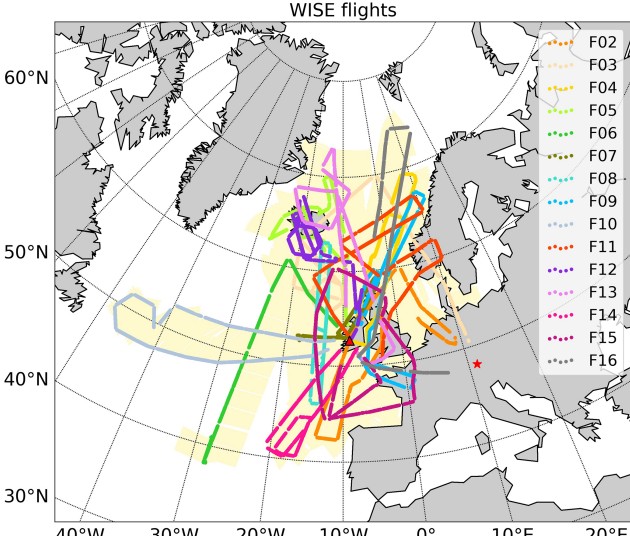

**Figure 1.** Overview of the 15 scientific flights of the WISE campaign. Color points correspond to the positions of HALO with GLORIA measuring. The red star indicates Oberpfaffenhofen, Germany and the red triangle Shannon, Ireland. The shade in light yellow gives a reference of the area covered by the measurements, indicating the distance of the tangent altitude point, i.e., of the closest point of the LOS of the instrument to the surface.

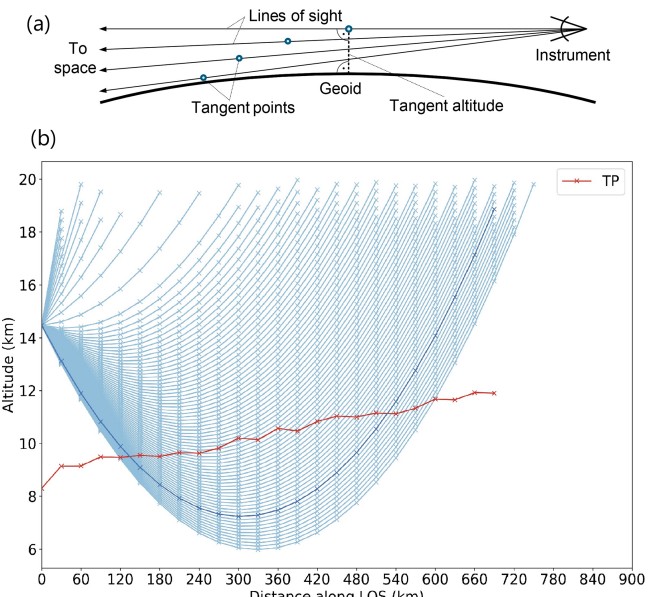

**Figure 2. (a)** Example of the measuring geometry of GLORIA. The tangent altitude point is the closest point of the LOS (line of sight) to the surface. **(b)** Example of the measuring geometry of GLORIA in a Cartesian system. Each blue line represents a different LOS that corresponds to a different elevation angle. The LOS is not a perfect straight line due to the refraction in the atmosphere, and when plotted in a reference system with the Earth's surface as a straight line, it adopts a parabolic form. Each LOS is associated to a different tangent altitude. The horizontal distance of each LOS can extend several hundreds of kilometers. The lowest LOS is the one with the largest path. The radiance measured by GLORIA is integrated along each LOS and contains the information related to the presence of clouds. The radiance from all LOS are recorded simultaneously. The red line indicates the tropopause (TP) from ERA5 along the corresponding LOS, in dark blue.

grated along the LOS of the instrument, the same applies for the IWC, thus the final parameter used for the comparison was the limb ice water path (IWP), i.e., the IWC integrated along the LOS (Spang et al., 2015). In addition, we retrieved the potential vorticity (PV) and equivalent latitude from the ECMWF data at the tangent point. The static stability ($N^2$) used to analyze the stability of the atmosphere was computed from GLORIA retrievals. The static stability is the square of the Brunt–Väisälä frequency ($N$), defined as

$$N = \sqrt{\frac{g}{\theta}\frac{\partial \theta}{\partial z}}, \tag{1}$$

where $g$ is the local acceleration of gravity, $\theta$ is the potential temperature and $z$ is the altitude of the air parcel. $N^2$ describes the vertical temperature stratification of the atmosphere and gives insight into whether an air parcel is in a transition region between the troposphere, characterized by low $N^2$ ($N^2 \approx 1 \times 10^{-4}\,\mathrm{s}^{-2}$), and the stratosphere, characterized by high $N^2$ ($N^2 \approx 5 \times 10^{-4}\,\mathrm{s}^{-2}$) (Grise et al., 2010).

The potential temperature product needed for the computation of $N^2$ was computed from pressure and temperature of the final dataset, which contains the retrieval results and a priori information taken from ECMWF. The results are dominated by a priori in regions where no measurements are available, i.e., in or below thick clouds.

## 2.4 The model: JURASSIC2

The JUelich Rapid Spectral Simulation Code V2 (JURASSIC2) is a fast radiative transfer model developed at Forschungszentrum Jülich for analyzing the measurements of remote sensing instruments (Hoffmann et al., 2008; Griessbach et al., 2013; Ungermann et al., 2015). It combines a forward model with retrieval techniques (for both limb and nadir geometries) and allows us to derive pressure, temperature and trace gas volume mixing ratios among others. JURASSIC2 solves the Schwarzschild equation (Petty, 2006; Wallace and Hobbs, 2006) in the mid-IR region using spectrally averaged radiances, the Curtis–Godson approximation (CGA; Curtis, 1952; Godson, 1953) and emissivity growth approximation (EGA; Weinreb and Neuendorffer, 1973; Gordley and Russell, 1981) in combination with emissivity look up tables (LUTs) (Ungermann et al., 2011). The LUTs are typically computed by the line-by-line Reference Forward Model (Dudhia, 2017).

JURASSIC2, together with the Juelich Tomographic Inversion Library (JUTIL), generates the level 2 products (temperature, trace gases, extinction coefficient). For a detailed description of this process the reader is referred to Ungermann et al. (2015). For this study, the level 2 product used is simply the extinction coefficient. An explanation of how it is retrieved is given in Sect. 3.2.

## 3  Cloud detection methods

To analyze the data, two methods to identify optically and vertically thin clouds at high altitudes were used. One method used the cloud index and the other the extinction coefficient.

### 3.1  Cloud index

The cloud index (CI) was first introduced by Spang et al. (2001) and has been widely used in different studies for the analysis of clouds in the UTLS and polar stratospheric clouds observed by CRISTA and MIPAS (Sembhi et al., 2012; Spang et al., 2015, 2016). The CI is a dimensionless number defined as the ratio between the mean radiances of two microwindows:

$$CI = \frac{\overline{I}_1\left(\left[788.2, 796.2\,\mathrm{cm}^{-1}\right]\right)}{\overline{I}_2\left(\left[832.4, 834.4\,\mathrm{cm}^{-1}\right]\right)}. \tag{2}$$

The first spectral window is mainly dominated by emissions of a $CO_2$ Q-branch and the second is an atmospheric window region. The CI is affected by the water vapor continuum contribution to the atmospheric window at low altitudes and depends slightly on latitude and season (Sembhi et al., 2012). When clouds are present, the emission in both microwindows increases. However, the relative increase in the $CO_2$ Q-branch is smaller. As a result, the ratio decreases and therefore a low CI indicates cloudy conditions. A $\sim 1.1 < CI < 4$ indicates the presence of clouds (Spang et al., 2008, 2015).

### 3.2  Extinction coefficient retrieval

The extinction coefficient (from now on simply extinction) was retrieved with JURASSIC2. Although scattering by cloud particles has an impact on the measured radiance (Höpfner and Emde, 2005), we simulated the radiative transfer without scattering. As explained by Höpfner and Emde (2005) the difference between zero scattering and multiple scattering for a case that falls between the two cases presented in their study ($\omega_0 = 0.24$ and $\omega_0 = 0.84$) would be between 25 %–28 %. Additionally, we performed test retrievals with single scattering for two flights using the radiative transfer model JURASSIC2. These flights were selected because both thin cirrus and thick cirrus were observed and, therefore, constitute an interesting case for studying the influence of scattering in different cases. The difference (calculated as

the mean of the median difference of both flights) between the extinction neglecting scattering and the extinction including single scattering is 21 %, with 73 % as the 95th percentile and $-86\%$ as the 5th percentile (results not shown). The mean difference at $2\sigma$, i.e., the 16th percentile and 84th percentile is $-4\%$ and 49 %, respectively. For the retrievals with single scattering, the CTH was computed following the same procedure as for the retrievals with no scattering. For 98 % of all cases, both retrievals detected a cloud. The altitude of the CTH was for about 71 % of the detected clouds the same, being the typical difference 0–0.375 km. Following the same procedure for the CBH, for 90 % of all cases, both retrievals detected a CBH, obtaining the same altitude in 58 % of the coincidental profiles. The typical difference for the CBH was also 0–0.375 km. The detected clouds for both flights and both retrievals covered the range 45–75° N, with the largest occurrence between 55–75° N. In consideration of these results, we conclude that for our current purpose of obtaining macro-physical properties of cirrus clouds the non-scattering approach is sufficient.

Obtaining the extinction means solving an ill-posed inverse problem. In our inverse problem, there is a state vector $\boldsymbol{x}$ describing the state of the atmosphere (quantities to be retrieved), a measurement vector $\boldsymbol{y}$ with error $\boldsymbol{\epsilon}$ and a forward model $\boldsymbol{F}$ implementing the physics of the involved processes.

$$\boldsymbol{y} = \boldsymbol{F}(\boldsymbol{x}) + \boldsymbol{\epsilon} \tag{3}$$

For this work, $\boldsymbol{x}$ is the extinction and $\boldsymbol{y}$ is the radiance in the microwindow 832.4–834.4 $\mathrm{cm}^{-1}$. This interval is the same one used for the CI. For a detailed explanation of the retrieval the reader is referred to Ungermann et al. (2015).

The retrieval grid consists of a constant altitude grid with 81 levels ranging from 6 to 16 km with a sampling distance of 0.125 km. The model includes corrections of the tangent altitudes due to the elevation angle offset and the refraction. Several tests comparing the radiance of a theoretical case of a cloud as a step function and the retrieved one were performed to determine the influence of the radiance of cloudy pixels on the pixels above (not shown), i.e., the effect of the PSF. The results show that the retrieved profiles are affected by Gibbs oscillations that cause ringing artifacts at the edges and an overshoot of $\approx 10\%$ is found (i.e., radiance larger than the maximum of the step function). These effects can cause an error in the determination of the cloud top height of one grid point ($\pm 125$ m). These oscillations could also affect the determination of the cloud bottom, creating a false detection of a thin layer (1–2 grid points) above a thick cloud in $\approx 1\%$ of all the cloudy profiles. The leading error term in the determination of the cloud top altitude is the pointing knowledge along the LOS. This error is about a tenth of a degree, which was validated by measurements of the Moon during several flights (Ungermann, 2021).

The range of retrievable cloud extinctions is from about $2 \times 10^{-4}$ to $4 \times 10^{-2}$ $\mathrm{km}^{-1}$ and allows for the detection of optically thin cirrus, one of the objectives of this study. The

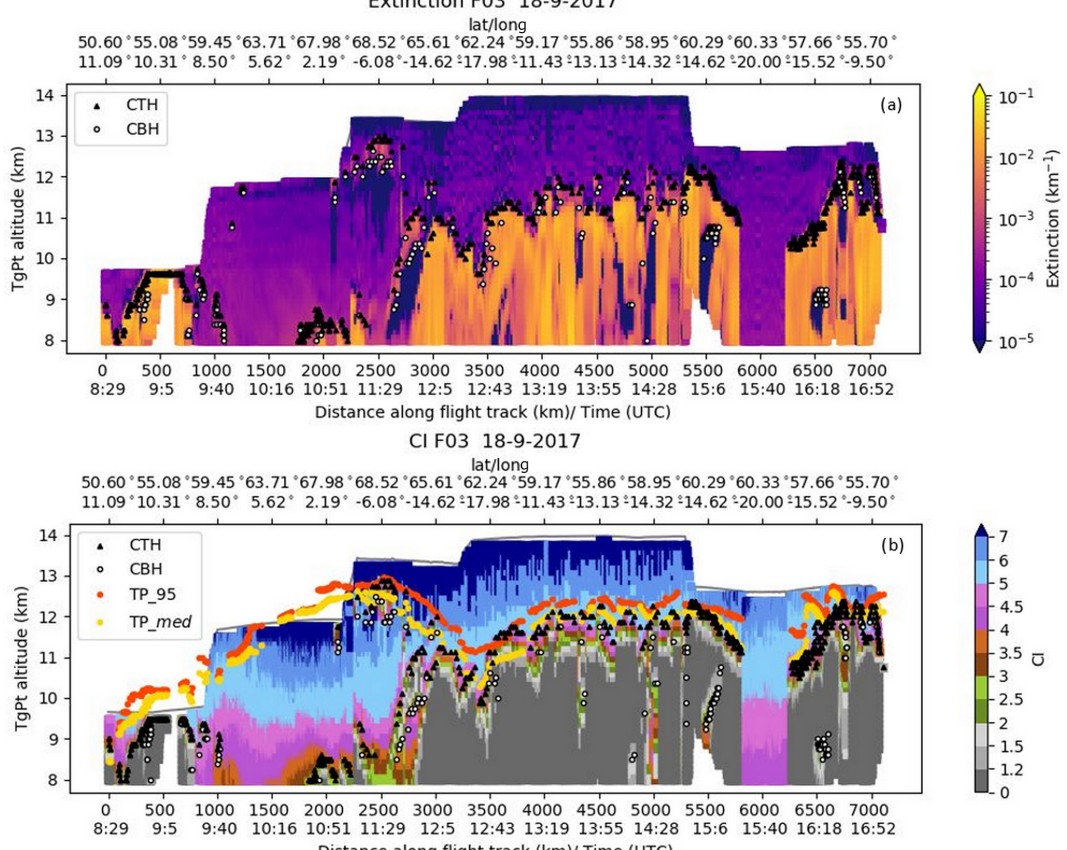

**Figure 3.** Cross sections of extinction **(a)** and cloud index **(b)** for flight 3 of the WISE campaign. The results are restricted to levels below the flight path. **(a)** Color code for extinction in $km^{-1}$. Orange-pink colors indicate the presence of clouds. **(b)** Color code for CI. Depending on the altitude, CI below 2 to 5 (colors from gray to pink) indicate the presence of clouds. Median tropopause ($TP_{med}$) and the 95th percentile of the tropopause ($TP_{95}$) are represented with orange and yellow circles, respectively. Cloud top height (CTH) and cloud bottom height (CBH) are represented with a black triangle and with a white circle, respectively. The altitude of the tangent points (TgPt) is the $y$ axis. The white areas in both cross sections correspond to a first filtering of optically thicker regions (CI < 2). These areas correspond to the tangent layers where the clouds are optically too thick.

upper limit is determined by the optical thick conditions in the limb direction and the lower limit by background aerosol and calibration uncertainties.

We sampled the CI on the extinction retrieval grid to allow for a comparison of both methods (Fig. 3). The radiative transfer model assumes for practical reasons a horizontal homogeneous atmosphere. As such, it assumes that simulated measurement rays diving below a cloud layer passes through the cloud twice, whereas in the actual situation it may "miss" the cloud on both occasions; if this occurs, the retrieval assigns nonphysical low extinction close to 0 to those regions (Fig. 3a, e.g., at 11:29 UTC, 11–12 km). Above the clouds (0.125–0.250 km), the low extinction is due to the second-order regularization that smooths the profiles and causes Gibbs oscillations in the extinction profile at strong extinction changes. For the CI cross section (Fig. 3b), depending on the altitude, different CI threshold values indi-

cate the presence of clouds. A detailed explanation about the detection threshold is found in Sect. 3.3.

### 3.3   Detection threshold for CI and extinction

To identify clouds in the measurements, we defined the detection thresholds for CI and extinction. First, we defined the criteria for clear sky regions. As a first approximation of clear sky conditions, profiles with CI always greater than 2 and extinction always less than $10^{-3}\,km^{-1}$ were selected. From this first coarse pre-selection, the vertical extinction gradient (Fig. 4) was computed to have an automated method that is more sensitive to optically thin clouds. If this gradient has a small variability that means there are no elements, i.e., aerosols or cloud particles, that cause a sudden increase in the extinction and therefore a large gradient. Clear sky profiles were defined to be those with an extinction gradient lower than a threshold defined as the median extinction

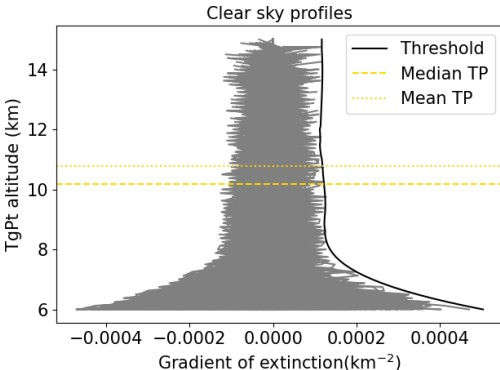

**Figure 4.** Vertical gradient of the extinction coefficient in the case of clear sky conditions. In black, the threshold defined as the median of the extinction gradient plus $5\sigma$. Altitude of the tangent points (TgPt) in the $y$ axis. The median tropopause height (Median TP) and the mean tropopause (Mean TP) height of all clear profiles is indicated by a dashed and a point yellow line, respectively.

gradient of the pre-selected profiles of all flights together plus $5\sigma$. $5\sigma$ was chosen after a visual fit to the gradient to reduce the number of false detections to a minimum. It is possible that the aircraft flies inside a cloud, which causes the vertical gradient of the extinction to be approximately constant and thus considered as clear sky. To exclude these cases, the condition that the CI must always be greater than 2 was added. Below 8 km the extinction gradient increases, which indicates the influence of the water vapor continuum at low altitudes (Fig. 4). Therefore, the analysis was limited to the range from 8 km to the aircraft altitude. For all clear sky profiles, probability density functions (PDFs) of CI and extinction were calculated and normalized for each altitude bin. Using the PDF for guidance, a threshold for each parameter was defined. The extinction coefficient threshold ($k_{thres}$) was defined as the median of the extinction plus $5\sigma$. This threshold is sensitive to structures with very low extinction, down to $2 \times 10^{-4}$ km$^{-1}$ for a tangent point between $\sim 11.5$ and 15 km (Fig. 5a, b). This detection limit is similar to the one provided by Sembhi et al. (2012) for MIPAS, with an extinction detection limit above 13 km of $10^{-4}$ km$^{-1}$ and to the findings of Griessbach et al. (2020), specified in Table 1 of the cited study. The CI threshold ($CI_{thres}$) is the 1st percentile (%) shifted by 0.3 (CI). Above 12 km we applied a constant CI of 5 because the low number of observations and occurrences of clear sky shifts the threshold towards CI numbers that are too high. Our threshold for this and lower altitudes agrees with the one defined by Sembhi et al. (2012) for northern mid-latitudes and Spang et al. (2012) for the MIPAS instrument. The threshold lines separate the clear air and cloudy cluster from each other, following the vertical gradient of the clear air cluster (Fig. 5a, b). As seen in Fig. 5c, the relation between CI and extinction is not one-to-one. However, for a CI between 3 and 5, which corresponds to optically thin clouds (Spang et al., 2008), the relation is stronger.

### 3.4 Definition of the macro-physical characteristics

Here, we define the macro-physical characteristics of the detected cirrus clouds that are presented in Sect. 4: cloud top height (CTH), cloud bottom height (CBH) and their vertical extent. In the limb geometry, the position of the cloud along the LOS is not exactly known. For analyzing the data, the clouds were referred to the tangent point, i.e., the point of the LOS closest to the Earth's surface and the corresponding tangent height layer. Using this definition of the position of the cloud, the CTH was defined as the first point in which the extinction (or CI) is equal to or larger than the $k_{thres}$ (or less than or equal to the $CI_{thres}$). For the analysis, we assumed a homogeneous cloud layer, which may underestimate the real extinction. This could cause an underestimation of the CTH for some cases, in which the cloud is on the ray path far from the tangent point location (Kent et al., 1997). All CTHs belong to the first cloud detected, i.e., the analysis did not include multi-layer clouds (two or more clouds with a clear separation in between). The CBH of a cloud using the extinction method is the altitude of the first detection in the series of limb observations with an extinction smaller than the $k_{thres}$; this ensures the identification of an altitude at or below the true cloud bottom. For the CI method, the CBH was computed using the minimum of the CI gradient of the profile (Kalicinsky et al., 2021). CBH could only be reliably determined for optically thin clouds. For optically thick conditions, the CI profiles saturate and the extinction profiles decrease in an unrealistic manner. Optically thick profiles are characterized by CI lower than 1.2 from an altitude $h$ down to the lowest altitude (Spang et al., 2015, 2016). Optically thin profiles, i.e., with small extinction, are those for which it was possible to define a CBH. Figure 6 shows an example of a saturated CI profile and a profile for a cloud layer. It is possible to observe how the CI profile reaches saturation for CIs smaller than 1.2. The last macro-physical characteristic that was analyzed is the vertical extent, defined as the CTH–CBH.

### 3.5 Differentiation between clouds and aerosol

Enhanced aerosol number densities can also affect the CI and cause false cloud detection. To investigate if the presence of aerosol particles influenced the results, methods described by Griessbach et al. (2014) and Griessbach et al. (2016) were applied. These methods use the different wavelength dependence of ice and aerosols, such as volcanic ash or sulfuric acid, in five wavelength regions to establish thresholds that differentiate them. The results (not shown), indicated very little influence of these aerosols in our measurements.

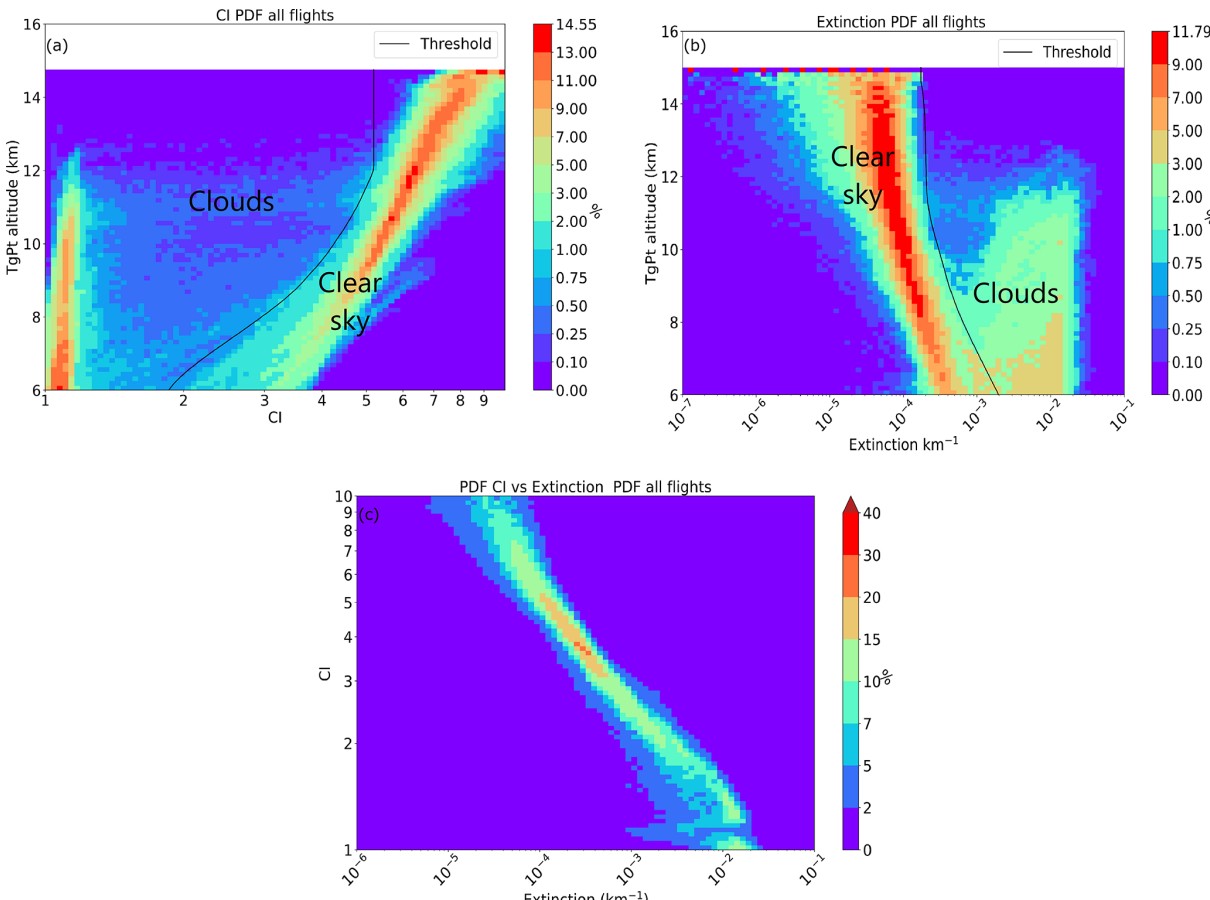

**Figure 5.** PDF for CI and extinction for all flights including all profiles. The bins are normalized by altitude. In black the threshold for differentiating cloudy conditions from clear sky. For panel **(a)**, clouds correspond to small CI, i.e, the left side of the $CI_{thres}$. For panel **(b)**, clouds correspond to high extinction, i.e., to the right side of the $k_{thres}$. The altitude of the tangent points (TgPt) is the $y$ axis. **(c)** PDF of CI as a function of extinction, normalized by CI. The total number of analyzed profiles is 13 539.

## 4   Results

### 4.1   Analysis: cloud top height and cloud bottom height

During the WISE campaign, 61 % of all observed profiles show CTHs using the extinction method and 59 % for the CI. 58 % of all profiles show a CTH for both methods, which indicates a similar performance. These fractions are comparable to the climatology presented in Goldfarb et al. (2001) for lidar observations, with a cirrus occurrence frequency of 60 % for fall. However, a fraction of 60 % is considerably larger than the ≈ 17 % reported by Sassen et al. (2008) for CALIPSO measurements and the International Satellite Cloud Climatology Project (ISCCP) for mid-latitudes. It is rather unlikely that this difference is related only to the disparate observational periods. We rather explain it by the differences in cirrus cloud selection criteria of the studies. While in our study there is no temperature threshold, Sassen et al. (2008) considered as cirrus only clouds with $\tau < \sim 3.0$–4.0 and with a maximum cloud top temperature of $-40\,°C$.

Goldfarb et al. (2001) considered for the detection of cirrus a threshold that was defined for each nightly determination and required that the cloud layer was in an air mass with a temperature of $-25\,°C$ or lower.

The extinction method and the CI method show good agreement in the determination of the CTHs, presenting a similar distribution (Fig. 7a, b). The CTHs between 8 and 10 km were observed in air masses with equivalent latitudes that spread from tropical to polar regions, having a slightly higher frequency at the polar latitudes. CTHs between 10 and about 12.5 km often occurred at equivalent latitude typical for mid-latitudes, whereas the CTHs above about 12.5 km were related to subtropical latitudes. The main difference between both methods is that the CTHs inferred from the CI are slightly higher (1–2 grid points) than for the extinction method.

From all considered profiles (13 539), about 39 % (5232 profiles) can be characterized as optically thick using the extinction method and 41 % (5517 profiles) the CI method; 36 % of all profiles are optically thick for both methods. The

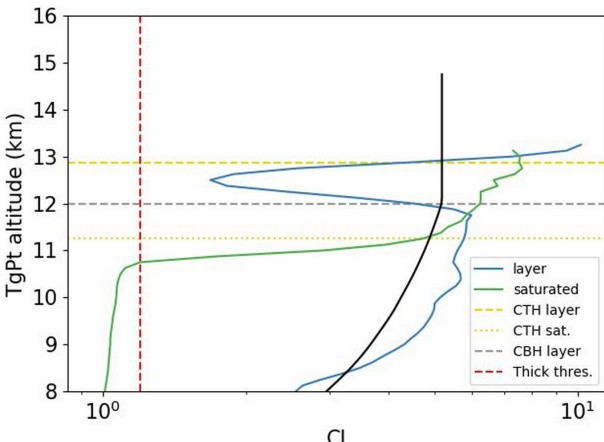

**Figure 6.** CI profile for a cloud layer (blue) and an optically thick case (green) that saturates. The horizontal yellow lines indicate the cloud top height (CTH) for the layer (dashed line) and the thick case (points line). The horizontal dashed line in gray is the cloud bottom height (CBH) of the cirrus layer. The red vertical line corresponds to CI = 1.2, i.e., optically thick cases. In black, the CI threshold. The altitude of the tangent points (TgPt) is the $y$ axis.

maximum extinction detected for thin clouds, in which a CBH was possible to determine, is $4 \times 10^{-2}\,\mathrm{km}^{-1}$.

The distribution of the vertical extent of clouds is presented in Fig. 8. The extinction method results in a higher amount of vertically thin clouds than the CI method, due to the slightly higher CBHs of the extinction method (Ungermann et al., 2020). For both methods, a large fraction of the optically thin cirrus clouds were located between 45–65° N and had a vertical extent smaller than 1.5 km (31 % of the clouds detected with the extinction method and 20 % of the clouds detected with the CI method). These results are qualitatively similar to the findings of Noël and Haeffelin (2007). They showed that between May and November the frequency distribution of the vertical extent of the observed clouds was biased towards values between 0 and 1.5 km. Our results are also in agreement with the mean layer thickness of 1.4 km found by Goldfarb et al. (2001).

## 4.2 Cloud top position with respect to the tropopause

The occurrence frequency of cirrus clouds above the tropopause remains a matter of debate. The vertical resolution of the underlying temperature profile of the meteorological analysis for the tropopause computation is a key point for respective analyses. As discussed in Pan and Munchak (2011), different definitions of the tropopause can lead to different results. For this study, the first thermal tropopause altitude was computed from ERA5 data. The LOS of GLORIA typically extends several hundreds of kilometers, hence the sampled air masses could be heterogenous in the horizontal. Further, the tropopause height was not constant along the LOS (Fig. 2). Two methods were used for representa-

tive tropopause definition for the air mass sensed by the instrument: (a) the median of the tropopause along the corresponding LOS of the CTH and (b) the 95th percentile. All extinction and CI cross sections of the WISE campaign with CTHs, CBHs, median tropopause ($TP_{med}$) and 95th percentile ($TP_{95}$) can be found in the Supplement. Figure 3 illustrates the case of a flight with both homogeneous and heterogeneous air masses. For example, the air mass at 16:18 UTC was homogeneous and $TP_{med}$ and $TP_{95}$ are close to each other (less than 125 m apart). At 11:29 UTC, there were heterogeneous air masses with $TP_{med}$ and $TP_{95}$ separated (three times the distance of the previous example), which affects the statistics of CTHs above the tropopause, since whether the CTH is located above or below the tropopause depends on the chosen tropopause altitude.

For the extinction method, the frequency of occurrence of CTHs above the $TP_{med}$ is 24 % of the total number of observations, whereas for the CI method the ratio is 27 % (Fig. 9b). The $\approx 3$ % difference is due to the CI detecting CTHs slightly higher than the extinction method. When considering $TP_{95}$, the percentages decrease to 13 % and 16 %, respectively, as it uses a more conservative criterion. This gives us confidence to conclude that CTHs above the lapse rate tropopause were detected, even when considering the error in the CTH determination, which is on the order of $\pm 125$ m. Figure 10 shows the distribution of all CTHs (Fig. 10a) and the distribution of CTHs above $TP_{med}$ (Fig. 10b) for the extinction method. As can be seen, most of the occurrences of CTHs above $TP_{med}$ were found between 50–70° N, with varying altitudes from 8–13 km. The few occurrences between 35–50° N were located at higher altitudes, from 10–14 km. About 6 % of all profiles show for both methods CTHs above the $TP_{med}$ and are classified as optically thin. The ratio of clouds with both CTH and CBH above the $TP_{med}$ is 2 % for the extinction method and 1 % for the CI method. When considering the $TP_{95}$, both occurrences decrease but CBHs above the TP were still detected. The presence of complete layers above the tropopause is inconclusive, since these CTHs and CBHs are only separated by one altitude bin and the CBH is only one or two altitude bins above the tropopause, which is within the uncertainties of the CBH. In Sect. 4.4, a potential case of a cloud layer above the tropopause is discussed in more detail. Our results (summarized in Table 2) agree with previous studies that claim the detection of CTHs above the tropopause for mid-latitudes. Goldfarb et al. (2001) used lidar ground-based instruments and found 5 % of CTHs at least 1 km above the tropopause and approximately 15 % above 0.5 km. Spang et al. (2015) analyzed CRISTA data and concluded to a 5 % frequency of occurrence of cirrus clouds (of all observations) and Zou et al. (2020) inferred their occurrence to 2 % for CALIPSO data and 4 %–5 % for MIPAS data. The analyses of Spang et al. (2015) and Zou et al. (2020) used the criterion of the cirrus CTH being 0.5 km above the ERA-Interim thermal tropopause. Using the same criterion, the frequency of occurrence is 4 % for

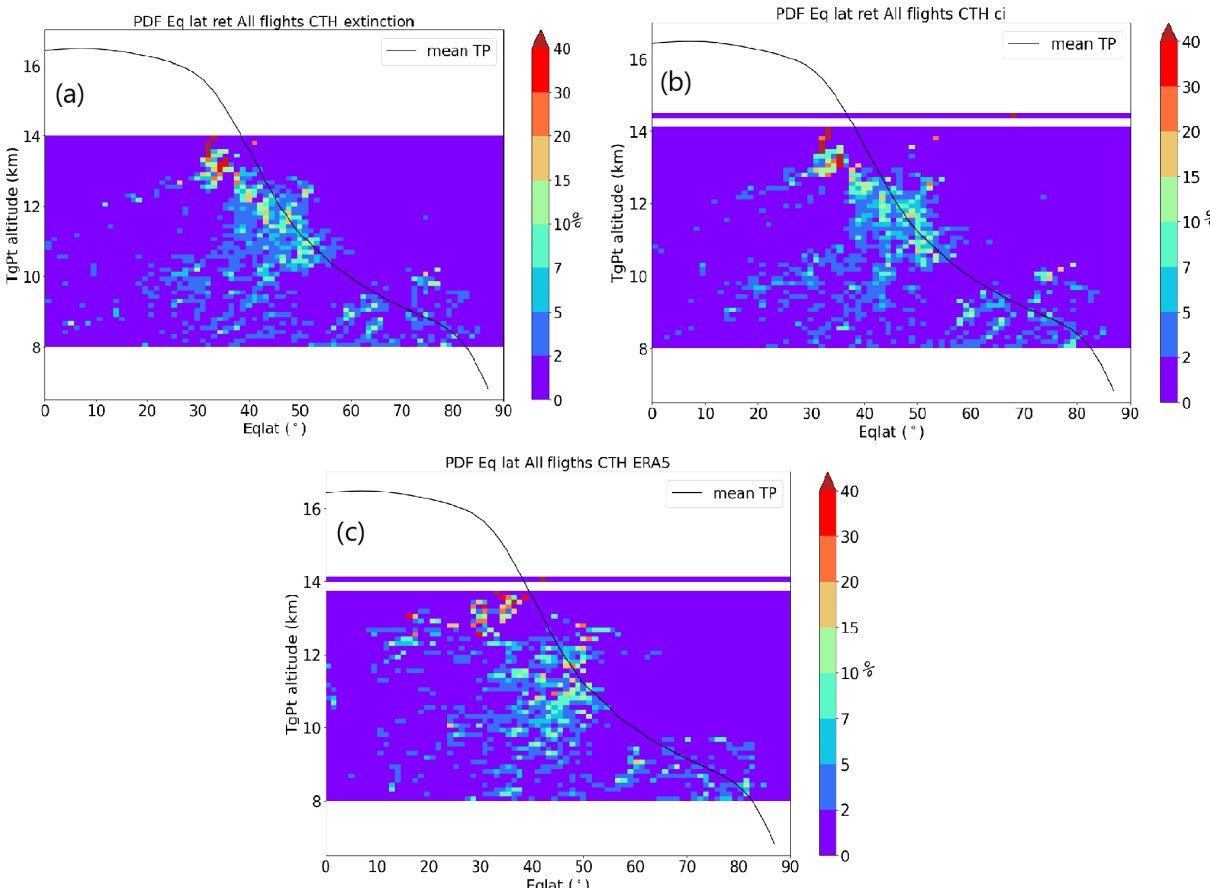

**Figure 7.** PDFs of CTH as function of equivalent latitude (Eqlat) normalized for each altitude bin from **(a)** the extinction, **(b)** the CI and **(c)** ERA5, as discussed in Sect. 4.3. The $y$ axis shows the altitude of the tangent points (TgPt). The black line represents the mean tropopause height during September and October 2017 as a function of the equivalent latitude. It was computed from ECMWF analysis data.

CTHs above the $TP_{med}$ for the extinction method and 7 % for the CI method. These occurrence frequencies are comparable to those reported in the literature (Goldfarb et al., 2001; Spang et al., 2015; Zou et al., 2020). However, as we used ERA5 data, which has a better vertical resolution than ERA-Interim, the equivalent criterion would be to mandate the cirrus CTH to be located 0.25 km above the tropopause. In this case, the frequency of occurrence increases to 13 % above the $TP_{med}$ for the extinction method and to 17 % for the CI method. We explain these differences in the frequency of occurrence by different periods being compared, the sensitivity and vertical resolution of the instruments, the uncertainty of the meteorological data used to estimate the tropopause height and the definition of stratospheric cirrus used in each study.

### 4.3   Comparison with ERA5

We compared our CTH detections with the ERA5 dataset by applying the observation geometry of GLORIA. As explained in Sect. 2.3, one of the parameters from ERA5 sam-

**Table 2.** Percentages of cloud top heights (CTHs) and cloud bottom heights (CBHs) detected above the median tropopause ($TP_{med}$) and the 95th percentile of the tropopause ($TP_{95}$) relative to all retrieved profiles for both detection methods. The last three rows correspond to the frequency of occurrence of stratospheric cirrus from the studies of Goldfarb et al. (2001), Spang et al. (2015) and Zou et al. (2020).

|  | $TP_{med}$ | $TP_{med}$ | $TP_{95}$ | $TP_{95}$ |
|---|---|---|---|---|
|  | CI | ext | CI | ext |
| CTH all | 27 | 24 | 16 | 13 |
| CTH thin | 7 | 7 | 5 | 4 |
| CTH and CBH | 1 | 2 | 1 | 1 |
| Goldfarb et al. (2001) | Lidar > 1 km | 5 |  |  |
|  | Lidar > 0.5 km | 15 |  |  |
| Spang et al. (2015) | CRISTA | 5 |  |  |
| Zou et al. (2020) | CALIPSO | 2 |  |  |
|  | MIPAS | 4–5 |  |  |

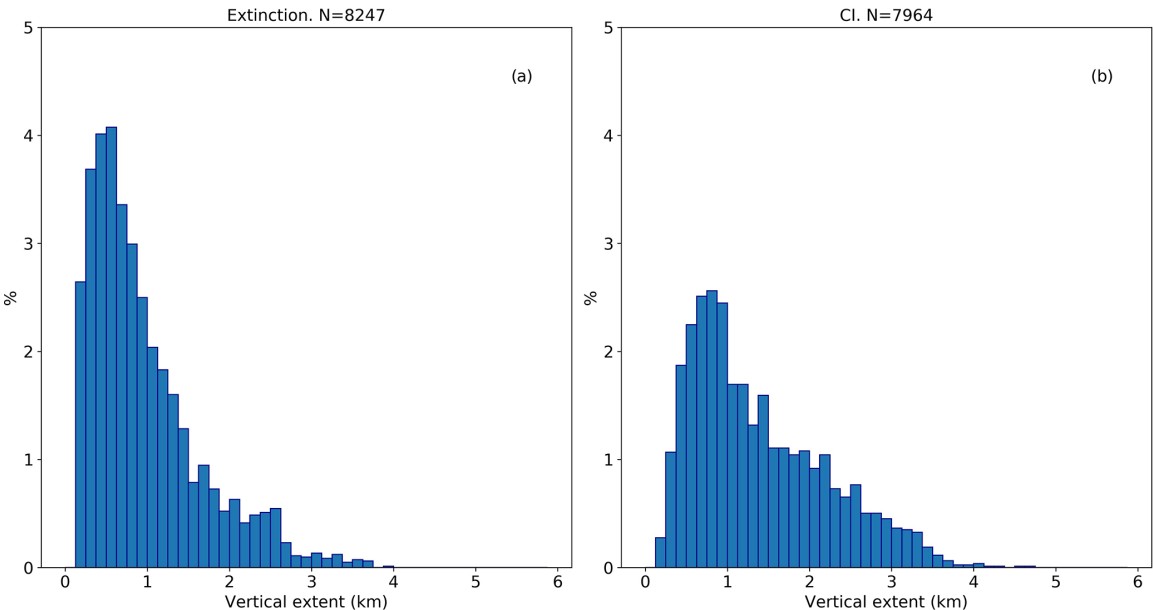

**Figure 8.** Distribution of the vertical extent of cirrus clouds for all flights for **(a)** the extinction method and **(b)** the CI method. The percentage is given in relation to the total number of CTHs ($N$) detected for each method.

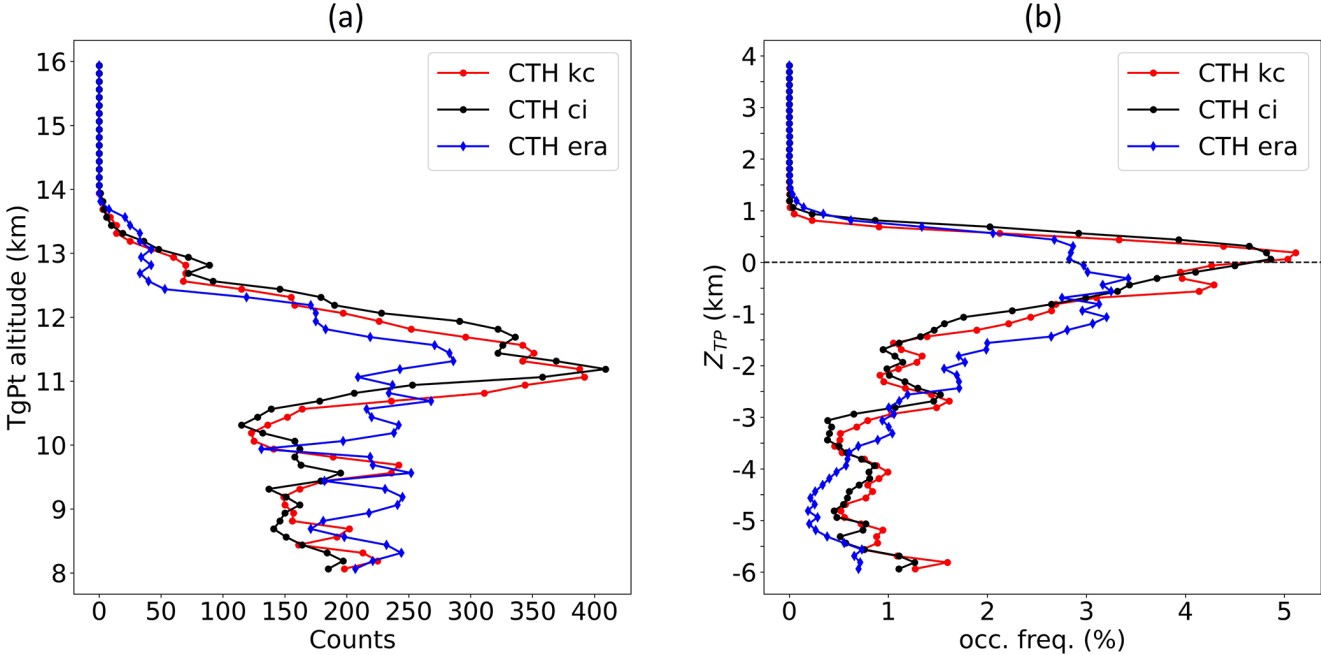

**Figure 9. (a)** Number of cloud top heights (CTHs) per altitude bin for the extinction method (kc) in red, the CI method (black) and ERA5 (blue). The altitude of the tangent points (TgPt) is the $y$ axis. **(b)** The same as panel **(a)** but using as coordinates the distance of the CTHs to the tropopause in km ($Z_{TP}$). The used tropopause is the median tropopause ($TP_{med}$). The three profiles were smoothed with a three points running mean.

pled following the viewing geometry of the GLORIA instrument is the IWC, which, when integrated along the LOS, results in the limb IWP. Spang et al. (2012) showed that CI and the limb IWP divided by the effective radius of the particles size distribution are very well related to each other. This is

since for large particles (with respect to the wavelength) the observed cloud radiances are determined by the integrated surface area along the LOS, in contrast to the volume density for small particles. We defined a CTH for the ERA5-based dataset to each tangent point with IWP > 0.

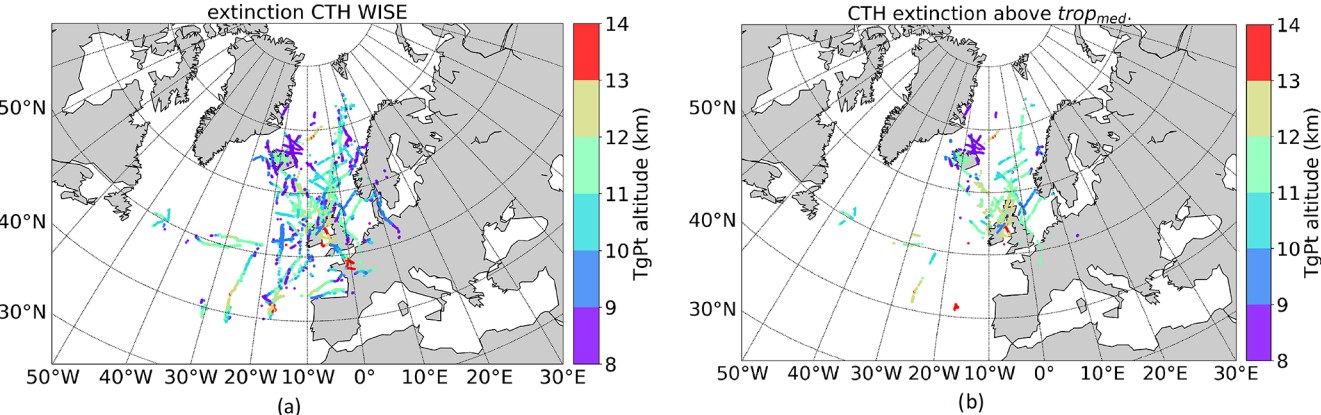

**Figure 10.** Distribution for **(a)** all cloud top heights (CTHs) for the extinction method and **(b)** CTHs for extinction method above the median tropopause (TP$_{med}$). Colors indicate the tangent point altitude (TgPt).

Figure 7c shows a similar pattern of CTHs inferred from ERA5 data as those derived from the measurements. From all investigated profiles, the fraction of detected CTHs is 59 % from ERA5, 59 % using the CI method and 61 % the extinction method. Figure 9a shows that between 8 and 11 km altitude, ERA5 indicates more frequent CTHs than the observations. This could be related to discarded multi-layer clouds in the detection algorithm, which could increase the number of CTHs observed between 8 and 11 km, as CTHs below the first CTH of the multi-layer cloud would also be included. The instrument is sensitive to higher and thinner cirrus clouds than the clouds assimilated by ERA5. Consequently, high CTHs detected by GLORIA will hide lower and thicker CTHs in the ERA5-based dataset. When changing to a coordinate system with respect to the TP$_{med}$ (Fig. 9b), the distribution of all CTHs is similar beyond 0.5 km distance from the TP$_{med}$. Between $-0.5$ and $0.5$ km there are more cirrus measured by GLORIA than present in ERA5. When considering all occurrences of cirrus above the TP$_{med}$, the observations indicate about 50 % more cirrus clouds than found in ERA5. This result indicates limitation in the cloud scheme used in the assimilation system of ERA5 for these optically thin clouds close to the tropopause.

## 4.4 Example of cirrus above the tropopause

The analysis presented in Sect. 4.2 suggests the presence of complete cirrus layers located above the tropopause, i.e., both CTH and CBH were found above the tropopause. As a case study, an observation made during flight 16 on 21 October was analyzed in more detail. Figure 11 shows a zoomed area of the cross section of the flight. For both the extinction method and CI, measurements with a cloud detection are marked by colors. The altitude for corresponding TP$_{med}$ and TP$_{95}$ are close, indicating that the sampled air masses were homogeneous with respect to the temperature structure around the tropopause. Both methods identify cirrus cloud

at 72.59 and 69.38° N with CTHs well above the tropopause ($\sim 0.5$ to 1 km for the first cirrus cloud and $\sim 0.5$ km for the second). For the extinction method, the CBH was located slightly higher than for the CI method, but still within the detection error. Therefore, the cirrus cannot unambiguously be ascribed to locations above the tropopause. At the location of the second cirrus there was a second tropopause at $\sim 18$ km. Therefore, the CTH of this cirrus was in between tropopauses. Both clouds were optically thin, with an extinction between $3 \times 10^{-4}$ and $5 \times 10^{-3}$ km$^{-1}$. The meteorological situation was characterized by a weak low-pressure system on the surface close to Iceland, with an occluded front. The clouds were located in an area where the wind at 200 hPa changed from southwest to west to northwest with velocities between 20–28 km h$^{-1}$. The air mass in both clouds had mid-latitude characteristics with an equivalent latitude of approximately 51° N. The cloud at 72.59° N was in an area where the PV varied from 2.4 to 6 PVU and was in a stable region with $N^2$ between 1.6 and $5.2 \times 10^{-4}$ s$^{-2}$. Therefore both the PV and $N^2$ indicate the transition region between troposphere and stratosphere (Kunz et al., 2009, 2011). The cloud at 69.38° N had a PV characteristic of stratospheric air masses between 3.7 and 5.7 PVU and large static stability, $5.6 < N^2 < 7.1 \times 10^{-4}$ s$^{-2}$. $N^2$ close to $7 \times 10^{-4}$ s$^{-2}$ is an indication of mixed sub-tropical and mid-latitudinal air masses (Kunz et al., 2009).

## 5 Conclusions

In this study, we analyzed cirrus cloud observations taken with the limb sounder GLORIA on board the research aircraft HALO during the WISE campaign. We used two methods for cloud identification, the cloud index and the retrieved extinction coefficient. The analysis focused on high cirrus clouds close to the tropopause and excluded multi-layer clouds. The extinction method indicated very thin clouds

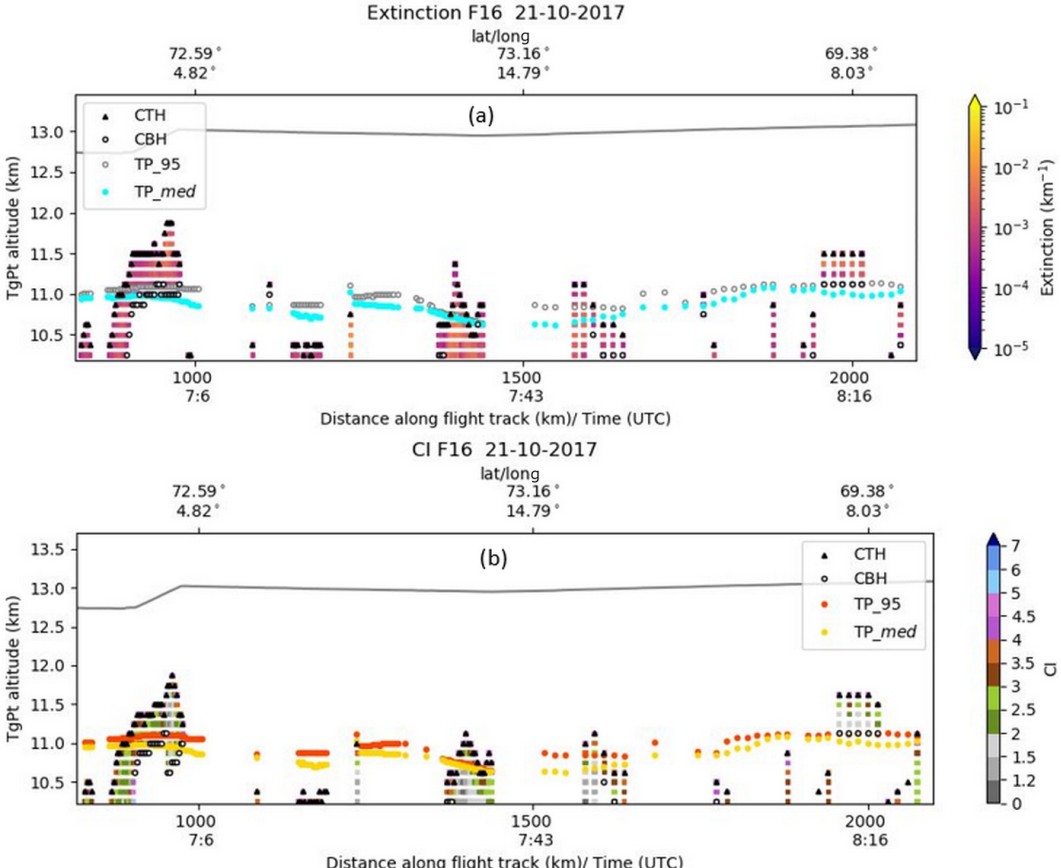

**Figure 11.** Zoomed area of the cross section of flight 16 on the 21 October 2017 focusing on two examples of thin cirrus above the tropopause. **(a)** Extinction coefficient color scale. Median tropopause ($TP_{med}$) and the 95th percentile of the tropopause ($TP_{95}$) are indicated with blue dots and gray dots, respectively. **(b)** Cloud index color scale. Median tropopause ($TP_{med}$) and the 95th percentile of the tropopause ($TP_{95}$) are indicated with yellow dots and orange dots, respectively. Black triangles indicate the CTHs and the white circles the CBHs. The gray line marks the flight trajectory. The altitude of the tangent points (TgPt) is the $y$ axis.

with an extinction as low as $2 \times 10^{-4}$ km$^{-1}$. Both methods are in good agreement, having similar frequencies of occurrence and similar CTHs. The main differences are the slightly higher CTHs of the CI method and the higher CBHs from the extinction method. For studying the presence of cloud tops above the tropopause we used two approaches. First, we calculated the median tropopause from ERA5 along the LOS of the GLORIA instrument and second, we used the more conservative 95th percentile. We considered similar tropopauses as an indication of homogeneous air masses. The frequency of occurrence above the tropopause varied from 27 % to 16 % for the CI and from 24 % to 13 % for the extinction method, where the differences between both approaches were due to LOS scenes with heterogeneous tropopause heights. Our results support the higher occurrence frequencies reported in the literature (Goldfarb et al., 2001; Spang et al., 2015; Zou et al., 2020) in contrast to lower frequencies derived from CALIPSO (Pan and Munchak, 2011; Zou et al., 2020) at midlatitudes. Using the same criterion as in Spang et al. (2015);

Zou et al. (2020), i.e., 0.5 km above the tropopause, the frequency of occurrence is 4 %–7 %. However, as the ERA5 dataset presents a higher vertical resolution, when analyzing the frequency of occurrence 0.250 km above the tropopause, the fraction increases to 13 %–17 %. This means that when the uncertainty of the tropopause estimate and the measurements is smaller, the stratospheric cirrus cloud occurrence frequencies are even higher. At 1.5 km below the tropopause, both identification methods present good agreement with the clouds indicated by the ERA5 dataset, when taking the observation geometry of GLORIA into account. However, the observed occurrence of cloud tops close to and above the tropopause is about 50 % higher than indicated by ERA5. We found CBHs above the tropopause, but they were within the uncertainties. Consequently, the GLORIA WISE campaign data cannot confirm the presence of unattached cirrus layers above the first thermal tropopause, but can confirm the presence of cirrus clouds at the tropopause with CTHs penetrating well into the lower stratosphere.

*Data availability.* The retrievals can be requested from the author.

*Supplement.* The supplement related to this article is available online at: https://doi.org/10.5194/amt-14-1-2021-supplement.

*Author contributions.* IB performed the analysis and wrote the paper with contributions from all authors. RS retrieved and resampled the ERA5 based dataset to the GLORIA geometry. JU prepared the GLORIA level 0 and level 1 data and supported IB in the setup of the JURASSIC2 simulations. SG, JU and MH contributed with their expertise in radiative transfer and SG also with her knowledge about the discrimination of ice particles from aerosols and cloud bottom altitude estimation. MK contributed with her expertise in cirrus. MR coordinated the WISE campaign and flight planning. All co-authors contributed to the discussion of the results, read the manuscript and suggested improvements.

*Competing interests.* The authors declare that they have no conflict of interests.

*Special issue statement.* This article is part of the special issue "WISE: Wave-driven isentropic exchange in the extratropical upper troposphere and lower stratosphere (ACP/AMT/WCD inter-journal SI)". It is not associated with a conference. TS2

*Acknowledgements.* The authors are grateful to the ECMWF for providing operational analysis, forecast and reanalysis data. Special thanks to the GLORIA team, including the technology institutes ZEA-1 and ZEA-2 at the Forschungszentrum Jülich and the Institute for Data Processing and Electronics at the Karlsruhe Institute of Technology. The authors also thank the WISE team, DLR-FX and the pilots.

*Financial support.* This research has been supported by the Deutsche Forschungsgemeinschaft (DFG) in the "Cirrus clouds in the extra-tropical tropopause and lowermost stratosphere region" (CiTroS) project (project no. SP 969/1-1, part of the HALO Priority Program SPP 1294).

The article processing charges for this open-access publication were covered by a Research Centre of the Helmholtz Association.

*Review statement.* This paper was edited by Andre Butz and reviewed by Klaus Pfeilsticker and two anonymous referees.

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

**Remarks from the typesetter**

TS1    Please note that the corrections of "numbers" are not language changes. Please give an explanation of why this needs to be changed. If you still insist on changing this values, the editor has to approve this change.

TS2    Please note that the SI was submitted and approved in this form, therefore, the title has to remain this way.