# Peer review of "Observation of Cirrus Clouds with GLORIA during the WISE Campaign: Detection Methods and Cirrus Characterization"

_Atmospheric Measurement Techniques, 2020_

## Referee Comment (RC1) · Klaus Pfeilsticker (Referee) · 22 Nov 2020

**Review of manuscript amt-2020-394 by Bartolome et al.,**

The paper reports on the detection of optically thin cirrus clouds employing the novel midIR limb sounder GLORIA (Gimballed Limb Observer for Radiance Imaging of the Atmosphere). The measurements were performed during the airborne WISE (Wave-driven ISentropic Exchange) campaign over the Northern Atlantic in September/October. For the retrieval of macro-physical characteristics of the cirrus cloud two methods are employed, namely the colour index (CI) method comparing mean radiances measured in two micro-windows $[I(788.2$ $cm^{-1})- I(796.2\ cm^{-1})]/[I(832.4\ cm^{-1} -I834.4\ cm^{-1})]$ and the extinction method, which employs a full (spectral) retrieval of the received radiances in the micro-window $832.4 – 834.4\ cm^{-1}$. Both methods have been tested in the past, and accordingly it is found that both methods compare well within the given error range to infer the targeted macro-physical parameters (i.e. cloud top and bottom height, the vertical extend and the cirrus cloud extinction). The measurements are also used to tackle the question, as to whether cirrus clouds may eventually exist just above the tropopause, but the reported findings may not support an unambiguous conclusion on this matter.

The paper is generally well structured and thought-through and the contents is certainly suitable to be published in AMT. Nevertheless, I recommend to revise the manuscript with respect to some issues, as they are listed below. Also, I strongly feel the manuscript would deserve some polishing with the English, as well as in removing some technical and typographical deficits. A (probably not complete) list of these deficits is provided below, including some recommendations for improvements.

**Comments**

1. Page, line 7: *We developed an optimized cloud detection method and derived macro-physical characteristics of the detected cirrus clouds such as cloud top height, cloud top bottom height, vertical extent and cloud top position with respect to the tropopause.*

This sentences needs a revision in order to reflect the full range of parameters (c.f., the cloud extinction inferred at 833 $cm^{-1}$), which are inferred in the study.

2. Since it is well known that the (midIR) extinction of cirrus clouds is (weakly) wavelength dependent, which itself is a function of the particle shape and distribution et cetera (e.g., Van de Hulst 1957, Yang et al., 2001, Baran, 2005, and others), you will need to address this issue in the introduction in a few sentences and evenly important as a consequence to restrict all statements in manuscript referring to the extinction to the considered midIR wavelength/wavenumber range.

Refs:
- Van de Hulst, H. C. 1957 *Light scattering by small particles*. Wiley, New York, USA
- Yang et al., Radiative properties of cirrus clouds in the infrared (8–13µm) spectral region, Journal of Quantitative Spectroscopy &Radiative Transfer 70 (2001) 473–504
- Baran AJ., The dependence of cirrus infrared radiative properties on ice crystal geometry and shape of the size-distribution function, Q. J. R. Meteorol. Soc. (2005), 131, pp. 1129–1142

3. Somewhere (page 2, lines 13 and 14) in the introduction, it needs to be mentioned, which definition of the tropopause is used in the manuscript (in agreement with your sentence on page 12, line 26 'As discussed in Pan and Munchak (2011) different definitions of the tropopause can lead to different results.').
4. Page 12, line 5: We rather explain it by the differences in cirrus cloud selection criteria of the studies. Isn't it also a matter of detection sensitivity for the two set of observations, and

if yes, how would the inferred cloud fractions compare for the same detection threshold of the extinctions?

**Technical, grammatical and typographical corrections**

Note: Some of the recommendations for corrections as listed below are optional others are mandatory but all of them are meant to improve the readability of the manuscript. The direct citation from the manuscript is given in *italic*.

1. Throughout the manuscript, I found the arbitrary change in tenses (c.f. from simple past, to presence and vice versa) rather irritating. Check for internal consistency and the appropriateness for the used tenses.
2. Page 1, line 8: What is a 'cloud top bottom height'?
3. Page 2, line 30 and elsewhere: I wonder whether the notation of a ‚value' is really needed to describe the magnitude of a physical quantity, c.f. *This IWC value matches*…., instead of …. This IWC matches; in the Figure 3 legend: *CI value* instead of CI; page 11, line 3: *CI-values lower than 1.2* ... → CI lower than 1.2…. page 12,line 3: *These values*…→ These fractions ..; page 16, line 24: The corresponding TPmed and TP95 have close values → The altitude for corresponding TPmed and TP95  are close  et cetera
4. Page 3, line 14. …. *of providing information* (of what?, c.f. on cirrus clouds) in the observation gap ….
5. Page 3, line 19: for measuring (what ?, c.f. microphysical parameters) of optically and vertically thin cirrus…..
6. Page 3, line 22: ….. horizontally averaged spectrum…. Mention here the size of the horizontal dimension/footprint over which it is averaged
7. Page 3, line 31: Table 1 summarizes the most important technical characteristics (→ features) of GLORIA
8. Page 4, Table 1 ledend: Observer altitude of 15 km and tangent altitude of 10 km. *Ungermann (2020, in preparation)* →  (Ungermann et al., 2020, in prep.)
9. Page 4, line 1: Provide a reference (Hoffmann, 2006) for the RT model JURASSIC2 here
10. Page 5, line 1: define 'ice water content (IWC)' on the first occurrence in the manuscript.
11. Page 5, line 3: In addition, we retrieved the potential vorticity (PV) and equivalent latitudes…for consistency put latitude in singular, or potential vorticity into plural.
12. Page 7, equation 1: Check for the correct notation
13. Page 7, line 18: … as the percentile 95 and -86% (correct?)
14. Page 8, line 5: The range of retrievable extinction values for clouds→ The range of retrievable clouds extinctions …
15. Page 8, line 20: *If this gradient has a small variability, that means there are no elements that cause a sudden increase in the extinction.* Please reformulated this sentence in order to make better clear what is meant.
16. Page 8, line 31: *This value is similar ….* → This detection limit is similar….
17. Page 8, line 34: *….. the low number of counts shifts (what counts?)…* → the low number of positive cirrus cloud detection ? ………..
18. Page 8, line 35: *This value, as well as the threshold for lower altitudes, agrees….* → Our threshold for this and lower altitudes, agrees
19. Page 9, line 6: *and vertical extent.* → and their vertical extent
20. Page 9; line 9: *which the extinction (or CI) has a value equal to or larger than the $k_{thres}$* → which the extinction (or CI) is equal to or larger than the $k_{thres}$
21. Page 10, line 4: *….first point with an extinction* (what point? and to what refers first?) → ….first detection in the series of limb observations with an extinction…
22. Page 11, line 4: *Thin profiles*…. What are thin profiles? Profiles of small extinctions? …
23. Page 11, line 6: *…..reaches saturation after CI = 1.2 …*(after?) → for CI's larger than 1.2.
24. Page 11, line 11: *…the different spectral slopes*..--> the different wavelength dependence…

25. Page 12, line 5: However, 60% is considerably…→ However, a fraction of 60% is considerably

26. Page 12, line 10: …….8 and 10 km present equivalent latitudes… → ….8 and 10 km as function of equivalent latitudes

27. Page 12, line 11: *For CTHs between 10km and about 12.5km the air masses have an equivalent latitude typical of mid-latitudes, whereas the highest CTHs, above about 12.5km are almost subtropical.*--> CTHs between 10km and about 12.5km often occur at equivalent latitude typical for mid-latitudes, whereas the CTHs above about 12.5 km, are (were) related to subtropical latitudes.

28. Page 12, line 13: *The main difference between both methods is the slightly higher (1 – 2 pixels) CTHs of the CI method.* → The main difference between both methods is the slightly higher (1 – 2 pixels) → CTHs inferred from the CI are slightly higher (1 – 2 pixels) than for the extinction method.

29. Page 12, line 14: *Considering all observed profiles about 39% are optically thick using the extinction and 41% the CI method.* → From all considered profiles, 39% can be characterized as optically thick (provide a number here) using the extinction method and 41% the CI method.

30. Page 12., line 14: *The maximum extinction detected for thin clouds in which a CBH was possible to determine is 4×10-2 km−1.* → For optically thin clouds, the maximum extinction at CBH was 4×10-2 km−1.

31. Page 12, line 20: *…the vertical extent distribution*→ the frequency distribution of the vertical extent

32. Page 12, line 22: … *computed*… → … found…

33. Page 12, line 27: … the first thermal tropopause altitude was computed from ERA5 data .. 'first' with respect to what?

34. Page 12, line 28: …. sampling air masses that can be heterogeneous. Consequently, the tropopause is usually not constant ..--> hence the sampled air masses can (could) be heterogeneous in the horizontal. Further, the tropopause height is (was) not constant.

35. Page 12, line 29: ….. *were applied*… → were used

36. Page 12, line 33: … *the air mass at 16:18 UTC is homogeneous*… → the air mass at 16:18 UTC was homogeneous (see my comment 1 above).

37. Legend 7: *PDFs of equivalent latitude (Eqlat) normalized for each altitude bin for (a) CTH detected with the extinction, (b) CTH detected with the CI and (c) CTH from ERA5. The altitude of the tangent points (TgPt) is the y axis* → PDFs of CTH as function of equivalent latitude (Eqlat) normalized for each altitude bin from (a) the extinction, (b) from the CI, and (c) from ERA5. The y axis shows the altitude of the tangent points (TgPt).

38. Page 13, line 1: … *there are heterogeneous*… → there were heterogeneous

39. Page 13, line 2: …. *since the CTH is above or below the tropopause depending on the chosen tropopause altitude.* → since as to whether the CTH is located above or below the tropopause depends on the chosen tropopause altitude. →

40. Page 13, line 11: …. both percentages decrease but still detect CBHs above the TP…→ both occurrences decrease but still CBHs above the TP are detected.

41. Page 13, line 12: *The presence of complete layers above the tropopause is inconclusive, as these CTHs and CBHs are in general just one altitude bin apart and the CBH is only one or two altitude bins above the tropopause, which is within the uncertainties of the CBH.* → The presence of complete layers above the tropopause is inconclusive, since for the cases CTHs and CBHs only separated by one altitude bin and the CBH is only one or two altitude bins above the tropopause, which is within the uncertainties of the CBH.

42. Page 14, line 5: *Spang et al. (2015) analysed CRISTA data (Spang et al., 2015) and concluded with a frequency of occurrence of 5% of all observations and Zou et al. (2020) obtained 2% for CALIPSO data and 4 – 5% for MIPAS data.* → Spang et al. (2015) analysed CRISTA data for cirrus clouds and concluded to a 5% their frequency of occurrence and Zou et al. (2020) inferred their occurrence to 2% for CALIPSO data and 4 – 5% for MIPAS data.

43. Page 14, line 7: …. *above the tropopause derived from ERA-Interim.*--> above the ERA-Interim thermal tropopause.

44. Page 14, line 9: *These values are comparable to the ones of the literature.* → These occurrence frequencies are comparable to those reported in the literature (provide references here).
45. Page 14, line 10: …. *ERA-Interim, the equivalent criterion would be 0.25 km above the tropopause.*--> ERA-Interim. Accordingly, an equivalent criterion would be to mandate the cirrus CTH to be located 0.25 km above the tropopause.
46. Page 14, line 12: *We explain the differences*… → We explain these differences…
47. Legend Figure 9: *The three profiles have been smoothed with a three points running mean.*--> The three profiles were smoothed with a three points running mean.
48. Table 2, legend: *Percentage with respect to all retrieved profiles of cloud top heights (CTHs) and cloud bottom heights (CBHs) detected above the median tropopause (TPmed) and the percentile 95 of the tropopause (TP95) for both detection methods.*--> Percentages of cloud top heights (CTHs) and cloud bottom heights (CBHs) detected above the median tropopause (TPmed) relative to all retrieved profiles and the percentile 95 for their occurrence above the tropopause (TP95) for both detection methods.
49. Page 16, lines 1 -3: *As explained in Sect. 2.3, one of the variables sampled following the viewing geometry of the GLORIA instrument is the IWC for ERA5, which when integrated along the LOS results in the limb IWP.*--> As explained in Sect. 2.3, one of the parameters sampled by GLORIA is IWP, which can be compared with the ERA5 reanalysis when the ERA 5 IWC is integrated along the LOS.
50. Page 16, line 5:  ... *this is caused by the fact that for large* …> this is since for large particles …
51. Page 16, line 9: *Figure 7c shows a similar distribution of CTHs in ERA5 data as the one derived from the measurements*  … --> Figure 7c shows a similar pattern of CTHs inferred from ERA5 data as those derived from the measurements.
52. Page 16, lines 9 – 11*: The fraction of CTHs detected in ERA5 is about 59% of all profiles, the same as the one of the CI method (59%) and only slightly lower than the fraction for the extinction method (61%).* → From all investigated profiles, the fraction of detected CTHs is 59% from ERA 5, 59% using the CI method, and 61% extinction method.
53. Page 16, line 12: …. *to not considering … which would mean increase the number of CTHs observed between 8 and 11 km* → discarding…. increases the number of CTHs incorrectly attributed to 8 and 11 km altitude range.
54. Page 16, line 17: …. *than for ERA5.* → than reanalysed in ERA5.
55. Page 16, line 17: *Considering all occurrences above the TP$_{med}$, the observations detect about 50% more than ERA5 data-set.* → When considering all occurrences of cirrus above the TP$_{med}$, the observations indicate 50% more cirrus clouds than found in ERA5.
56. Page 16, line 21: *In Sect. 4.2 the presence of complete layers above the tropopause was suggested,*… → The analysis presented in Sect. 4.2 suggest the presence of complete cirrus layers located above the tropopause.
57. Page 16, line 23: …. *Only cloudy points*… (what are cloudy points?) -> For both the extinction method and CI, measurements with a positive cloud detected are marked by colours.
58. Page 17, lines 1 – 3: *The CBH is slightly higher for the extinction method and above the tropopause, but still within the detection error, therefore, no affirmation of it being undoubtedly above the tropopause is made.* → For the extinction method, the CBH is located slightly higher than for the Ci method but still within the detection error. Therefore, the cirrus can't unambiguously be ascribed to locations above the tropopause.
59. Page 17, line 3: *In the location*… → At the location
60. Page 17, line 7: *These values of PV and N2 indicate*… → Therefore both the PV and N2 indicate…
61. Legend Figure 11: Describe the grey line (i.e. the flight trajectory).
62. Page 18, line 2: … *and the derived extinction* ..-> and the inferred extinction…
63. Page 18, line 3: … *and did not include*…→ and excluded
64. Page 18, line 5: …. *with an extinction of 2×10-4 km−1*….--> with an extinction as low as 2×10-4 km−1.

---

## Referee Comment (RC2) · Anonymous Referee #2 · 9 Dec 2020

Referee Report on "Observations of Cirrus Clouds with GLORIA during the WISE Campaign: Detection Methods and Cirrus Characterization", by Bartolome et al.

General Comments

This paper analyses and characterises cirrus cloud occurences observed during a suite of scientific flights observing the limb in the infrared range with the GLORIA spectrometer. This paper is written with care and provide an in-depth analysis of the characteristics of cirrus clouds using two different retrieval methods, based on the extinction coefficient and on a cloud index, respectively. Although I am not sure about the validity of the approach neglecting the scattering in the radiative model, the clear quantifica-

tion of the expected biases is highly valuable and provides to the reader the necessary insight to make an own judgement.

Hence, I recommende the publication of this paper, after addressing minor issues.

Specific comments

L. 8, p.1: "cloud top bottom height": Do the authors mean: "cloud bottom height"? Otherwise, what is the difference between "cloud top bottom height" and "vertical extent"?

L. 9, p.1: What do the authors mean by "fraction of cirrus clouds"? Is it an estimate of covered area referred to the global coverage? Is it restricted to some altitude range with respect to the tropopause level? Is there some time reference in terms of annual mean? The estimate of 13 to 27 % is different from what is mentioned in L. 18, p.1: is it a number derived from the GLORIA measurements?

L. 12, p.1: What do the authors mean by "unattached cirrus layers"?

L. 17, p.1: Do the authors mean "one or banks of small white flakes"?

L. 20, p.1: "due to the low temperature of their environment": I guess the scattering due to the presence of the cirrus is the main driver, and not the temperature: I guess an even cold, dry, cloud-free atmosphere is not equally absorbing than cirrus clouds in the same temperature conditions. Please clarify.

L. 21, p.1: "they influence the amount of solar radiative energy received": Is this statement not in contradiction with L. 19-20, p.1: "cirrus clouds are rather transparent to incoming solar radiation"?

L. 21-22, p.2: "2% of stratospheric cirrus ... 4-5% for MIPAS in middle latitudes": To which quantity do these percentages refer? To the global cloud occurrence frequency? on an annual basis?

L. 35, p.2: What do the authors mean by "long light of sight"?

L. 2, p.3: What do the authors mean by "nature of the cirrus" ? Do they mean the optically thin/thick or subvisible character, or the (microscopic) structure of the clouds, or something else ?

L. 14-20, p.3: There are some repetition of information provided in p.2. The authors might consider remove them.

L. 23, p.3: Except saving memory space, does such averaging present any advantage in better visualizing the vertical structure, by getting rid of the local variability ? Table 1, p.4: This table seems to provide properties of both detectors (Fourier-transform spectrometer and 2D detector). It might be important to specify which one is concerned by each property. For instance, are both instruments covering the same range 780-1400 cm-1 ?

Figure 1: The choice of map does not render very well the location of land, and specifically of countries of importance, like Ireland and UK. Has the colour code (e.g. with all range of blue colours at the level of the sea and oceans) any importance for the present study ? If yes, the meaning of the colour code and a colour map should be provided. If not, I suggest to change the choice of map to make the information of importance (I guess: the geolocation) clearer and more visible.

L. 4-5, p.5: It seems useful to give some more insight into what static stability is, by giving some value of the static stability for extreme cases and/or by giving some equations that would also be useful to explicit the temperature dependence mentioned in l. 6, p.5. The quantity N should be defined. This paragraph indicates that the the static stability is an important parameter, but it is not clear why, and for which purpose the stability of the atmosphere has to be computed.

L. 10, p.5: If possible, I suggest to add some reference in English providing a description of JURASSIC2 for the more general readership of the journal.

L.10, p.5 - l.5, p.6: In view of the very scarse documentation provide about JURAS-

SIC, I suggest to extend a little bit the description, and make it somewhat less cryptic. E.g., concerning the retrieval technique, what is the retrieved quantity, and from which quantity-ies ? If the inversion technique has some similarities with the one used for limb sounding measurements or any other techniques, this might be mentioned. A reference is needed about the Schwarzschild equation.

L. 11-12, p.6: What do the authors mean ? Cloud index and extinction coefficient are physical quantities, and not methods.

L. 16-17, p.7: "The difference (. . .) is 21%": please specify the metric used.

L. 12-19, p.7: From this paragraph, it appears that the present case is a rather un-favourable case with respect to the study by Höpfner and Emde (2005). From runs on test cases that are not described, a quite large difference is found between the retrieved extinction coefficient using no-scattering approximation, and the retrieved extinction coefficient using single scattering (which is a simplified case with respect to the actual multiple scattering case). In the no-scattering case, quite high values are found for the most extreme cases (P5 and P95). From all these data, it is difficult to convince (at least the non-expert) that a non-scattering approach is sufficient. The authors might provide the values of the difference at 2 x the standard deviation (P16 and P84) or in-terquartile values or another useful metric. They also should specify how the test cases were chosen (using different typical cirrus cloud configurations ? Was the distribution of the different kinds of situations representative for real atmospheric conditions at the considered latitude?). Finally, an important and more convincing argument would be to provide an estimate of the uncertainty on all target macrophysical quantities and on the detected cirrus cloud fraction, possibly in function of the latitude.

L. 27, p.7: What do the authors mean by "elevation angle offset" ? A reference to a paper about the instrument might be useful.

L. 1-2, p.8: Should this error of 125 m be added to GLORIA's 140 m-vertical resolution ? How does it affect GLORIA's ability to detected small-scale structures mentioned in

[Figure]

§2.1 ? Caption Figure 3: "First filtering of optically thicker regions": Do the authors refer to the aerosol contributions ? It would be useful to specify.

L. 12-13, p.8: "Above the clouds": please specify the vertical range affected by this regularization effect. Which method is supposed to be affected by this problem ?

L. 18-19, p.8: A reference is needed for this choice of criteria for clear sky conditions. "CI always greater than 2": What do the authors mean by "always" ?

L. 18-20, p.8: In which extend is this pre-selection coarse ? Does it miss cloudy conditions, does it identify too much clear-sky conditions as cloudy ones, or randomly fails to distinguish clear-sky and cloudy conditions ?

Caption Figure 4: "Clear sky profiles of the vertical gradient (...)" looks strange. Wouldn't it be better to write "vertical gradient of the extinction coefficient in the case of clear sky conditions" ?

Figure 4: I guess the reason why the width of the scattering cloud is quite constant down to ~8 km, and then rapidly increases with decreasing altitudes (linked afterward with the water vapour influence), is related to the presence of the tropopause level. The authors could usefully add some line representative for the tropopause level for the sake of clarity.

L. 23-24, p.8: Do the authors expect to miss many cirrus events by using this apparently very conservative criteria (following Fig. 4)? ' L. 28, p.8: PDF should be defined.

L. 30, p.8: Where is the value 2 x 10-4 km-1 coming from ? Figure 4 provides a value of ~1.1 x 10-4 km.

L. 31-32, p.8: It would be useful that the authors provide here the estimates by Sembhi et al (2012) and bu Griessbach et al. (2020).

L. 32-34, p.8: Where are the estimates of the CI threshold and considerations about "low number of count shifts" coming from ?
L. 34, p.8-L.1, p.9: Similar to comment on L. 31-32, p.8.

L.3, p.12: It would be interesting to mention the percentage of profiles showing cloud occurrence detected by both methods, to see in which extend these methods have similar performance, or can be considered as complementary. This information would usefully complete the distribution shown in Figure 7.

L. 10-11, p.12: Is there any possible confusion between the occurrence of cirrus clouds and of polar stratospheric clouds ?

L. 14, p.12: Same as comment on L.3, p.12.

Caption Figure 7: The authors might consider adding, for Fig. 7c, that this plot is discussed in Section 4.3.

Figure 8: If the distribution shows the number of estimated CTH, I guess it is the same distribution as the distribution of detected cloud occurrences. The integral of this histogram should be about 100%, since all not considered values (values < 0 or > 6 km) are unrealistic, thus supposed to be at worst marginal. However, a first estimate gives a total of ∼36% in the case of the extinction method and even significantly less in the case of the CI method. What is wrong ? The estimates in L. 18-19, p.12 (31% [20%] of the clouds detected by the extinction [CI] method) look consistent with Figure 8.

Figures in supplement, L. 5, p.4: The supplement includes 15 figures seemly aimed at providing the data for each individual flight. However, flight 1 is missing, and flight 3, already illustrated in Figure 3, is duplicated. Is it what the authors want ? Also, the last figure, i.e. Figure S15, corresponds to the 16th flight although a total number of 15 flight is mentioned in L.5, p.4. This should be corrected for the coherence.

Caption Figure 10: "all cloud top heights (CTHs) for the extinction method", for 10a and 10b; "with color code as equivalent latitude".

Technical corrections

L. 2, p.2: "their detection"

L. 21-22, p.2: "4-5% for MIPAS at middle latitudes"

L. 4, p.4: "the data (. . .) were. . ."

————————————————————

---

## Referee Comment (RC3) · Anonymous Referee #3 · 14 Dec 2020

In this paper, the authors describe the results of retrievals of ice cloud detections from limb measurements made using the GLORIA instrument aboard in-situ flights during the WISE campaigns. The results are interesting given the unusual (to me) approach, and definitely fit the scope of the AMT journal. The sensitivity to ice clouds with extremely low optical depths is very interesting. I have several comments that I hope would make the paper more accessible to readers interested in cloud detection techniques but not familiar with the constraints of limb measurements.

**Major comment**

My main concern with the paper is that while I'm familiar with many remote sensing

and in-situ detection techniques, and with the cloud climatologies they enabled, I had trouble interpreting the results presented in this paper and putting them in perspective against other retrievals. I can't believe I will be the only one. In the minor comments below I try to explain where I had the most trouble understanding.

I am also concerned about the uncertainty in the horizontal location of retrieved clouds associated with the retrieval technique, and I would like to see it discussed, especially when relating the altitudes of detected clouds with tropopause.

**Minor comments**

p.1, L. 15-17: "It is possible... veils" this is generic information, which has no bearing on the following discussion and actually creates confusion, as the reader might believe it needs to retain the difference between cirrus, cirrocumulus and cirrostratus to understand what follows. I think it can be omitted.

P.2, L.2: "its" => "their"

P.2, L.2: I don't understand what are the "typical operational weather satellites" that are equipped with nadir sounders? Please be more specific. To my knowledge "typical operational weather satellites" are geostationary and provide visible/infrared imagery with near-global coverage. "Nadir sounders" make me think of active sensors, such as radars and lidars, which are not found aboard typical operational weather satellites.

P.2, L.15: "Cloud and Aerosol Lidar" – this is not the meaning of the CALIOP acronym. Please fix.

P.2, L.15: "as Dessler (2009)" why mention this?

P. 2, L.16: "there is not enough evidence of clouds above the tropopause in mid-latitudes" not enough evidence for what? Please be aware that more recent papers look at cirrus clouds above tropopause using CALIPSO data: e.g. Dauhut et al. 2019 https://doi.org/10.5194/acp-20-3921-2020

P. 2, L.22: "LIDARs" => "lidars"

P. 2, L.24: If you are interested in SVC cover derived from CALIPSO, please be aware of Martins et al. 2010 https://doi.org/10.1029/2010JD014519, Reverdy et al. 2012 https://doi.org/10.5194/acp-12-12081-2012, or Wang et al. 2019 https://doi.org/10.1029/2018JD029845

P. 2, L.26: Again, which nadir viewing instruments?

P. 2, L.33: I understand that this is the resolution of a pixel looked on straight from the plane, i.e. in a plane normal to the direction of view. What is the volume of air described by the measurement in a straight line away from the plane, I.e. along the line of sight? Is the tangent altitude the point of lowest altitude along the line of sight? Please mention these precisions.

P. 3, L. 9: "step stone" => "stepping stone"?

P. 3, L. 23: "one profile": what is the vertical resolution of one such profile?

P. 6, Figure 2: I'm not overly familiar with measurements like GLORIA's, so please forgive me if I ask obvious questions. Here are a few things I think I've understood, that I don't think are explicitly stated before the figure, and that would help its understanding if they were:

- I understand each blue line in Figure 2 to be a different line of sight.

- I understand GLORIA scans successive lines of sight, each corresponding to a different angle with the horizontal, and each describing cloud information along the line of sight that corresponds to that angle and to a minimum altitude level.

- I understand that the cloud information (IWC) is integrated along the line of sight, and the obtained IWP used as cloud indicator. It follows that the instrument documents the vertical distribution of clouds across a very large horizontal distance that varies with altitude, eg for 7km ASL the horizontal distance described is 420km, for 15km it is

660km, for 18km it is 720km, etc (according to the lines of Figure 2).

- I understand that analysis of GLORIA data can tell a cloud is located along a given line of sight, which provides some insights into its spatial location. Each value of GLORIA's vertical cloud profiles is representative of a quite long path, that is longest at highest altitudes, and neither horizontal nor vertical: for instance, the lowest line of sight in Figure 2 will document cloud presence at ~12km ASL 60km away from the plane, at ~10km ASL 120km away, down to ~6km ASL 330km away, and then up again.

My understanding of those points sometimes comes from later sections of the text. If my understanding is correct, could you please find a way to include those points in the text before reaching Figure 2, or in the legend of Figure 2, to help readers unfamiliar with the approach?

Figure 2: Would it be possible to add a cloud to this figure and show next to it the cloud profile that would then be generated, with the final base and top altitudes? It would help interpretation by non-experts.

Figure 2: What happens when a cloud is present at high distances from the plane, beyond the tangent, and is picked up by a line-of-sight that shoots low? For instance, a cloud located 600km away from the plane at 14km ASL could appear in the lowest line of sight. In that case, would the cloud be attributed a ~6km altitude, ie a 8km error ? Is there a way to know whether that situation is frequent, i.e. to quantify the error in vertical distribution due to that uncertainty on the cloud position along the line of sight?

Section 2.1: This might be painfully obvious to the authors, but I'm afraid I'm not sure I understand in which direction the scan is occurring? Are the lines of sight always pointing the same way with respect to the plane? Is the instrument looking in a different direction every time the plane changes direction?

P. 9, Figure 3: Assuming I've understood correctly how to interpret GLORIA's measurements, does it mean that cloud base and top can be located very different distances

away from the plane? Could you comment on whether this might impact the retrieved vertical distribution somehow?

p. 9, L. 7: "the clouds were referred to the tangent point" : Is this assumption made for convenience in absence of useful information, or is there a reason to think it is realistic? Is there a way to verify how frequently this assumption is reasonable or not, and quantify what the related uncertainty means for the cloud profiles?

Figure 3 and others: The WISE flights seem to cover a very large latitude range, from polar to almost tropical regions. Given we don't know where are clouds along the LOS, and what we know about the direction the LOS are pointing to, and the variable flight paths followed by the plane, is there a way to document the horizontal area described by these results?

P. 10, Figure 5: could you add a figure showing the distribution of CI vs. Extinction? It could help quantify whether both variables are interchangeable.

P. 12, L 26: "as discussed in Pan and Munchak (2011)..." The definitions of the tropopause are important, but the geographic colocation of the cirrus and the retrieved tropopause altitude is important as well. Given it is not possible to tell where a detected cloud is along the line of sight, it means there is an uncertainty of several hundred kms on the horizontal location of the detected cloud, with the largest uncertainties at the highest altitudes (closest to the tropopause). Trying to compare the altitude of a cirrus with the altitude of the tropopause when both can be separated horizontally by 600km makes me nervous, especially for clouds which are close to the midlatitude-tropics boundary (where the tropopause can vary by several kms in a few hundred kms). Could you comment on the impact the cloud location uncertainty can have on colocating it with the tropopause, taking into account the latitude?

P. 13, Figure 7: Could you maybe plot the zonally averaged tropopause altitudeon these plots?

P. 14, Figure 8: Would it be possible to separate the distribution of cloud properties depending on whether the clouds were observed in tropical, midlat or polar latitudes? I would expect the vertical extents to vary with latitude.

P. 14, L.5-6: "Spang et al..." this has already been stated close to the introduction. Please avoid repetition.

P. 15, Figure 10: I'm not sure I understand what is to be learned from these figures. If I understand correctly, they show the spatial distribution of detected cloud tops. Are we supposed to compare the left figure that shows all CTH with the right figure that shows only the CTH above the tropopause, and infer the regions where CTH above tropopause occur most frequently? If that is the case: 1) why change the color code between both figures? It makes the comparison much harder. Is the color code showing another point of comparison, or another independent information? 2) Isn't there an easier way to represent the density of CTH above the tropopause relative to the total number of measurements? (Eg counting the number of cloud points in 1x1° or 2x2° boxes ?) 3) why isn't that discussed in the text? I don't understand how Figure 10 relates to what is discussed P. 13 lines 8-12.

P.16, L. 12: The already stated CALIPSO references (Martins and Reverdy) show a larger frequency of SVC and focus on the Tropics, it might make sense to refer to them.
* * *

---

## Author Comment (AC1) · 29 Jan 2021

**Final response to referee #1 comments on paper amt-2020-394**

Dear Dr. Pfeilsticker,

we would like to thank you for all the helpful and constructive comments that improve the content and readability of the manuscript.

We proceed to answer and address all questions and comments. The comments are in bold letters and the answers in normal format. The page and line indicated in the answers correspond to the new version of the manuscript. At the end of the comments the revised manuscript is attached, with the additions in blue and the deletions in red.

**Referee #1**

**Specific comments**

1. Page1, line 7: We developed an optimized cloud detection method and derived macrophysical characteristics of the detected cirrus clouds such as cloud top height, cloud top bottom height, vertical extent and cloud top position with respect to the tropopause. This sentences needs a revision in order to reflect the full range of parameters (c.f., the cloud extinction inferred at 833 cm-1), which are inferred in the study. This sentence has been changed to the following (Page 1, line 7-9):

"We developed an optimized cloud detection method, based on the cloud index and the extinction coefficient at the microwindow 832.4-834.4 cm-1. We derived macro-physical characteristics of the detected cirrus clouds such as cloud top height, cloud top bottom height, vertical extent and cloud top position with respect to the tropopause."

2. Since it is well known that the (midIR) extinction of cirrus clouds is (weakly) wavelength dependent, which itself is a function of the particle shape and distribution et cetera (e.g., Van de Hulst 1957, Yang et al., 2001, Baran, 2005, and others), you will need to address this issue in the introduction in a few sentences and evenly important as a consequence to restrict all statements in manuscript referring to the extinction to the considered midIR wavelength/wavenumber range.

This issue is now addressed in the introduction, which has been modified accordingly:

Page 3, line 3-5: "The mid-IR radiative properties of cirrus clouds depend on particle size, particle shape and the considered wavelength (Baran 2005, Hulst 1957, Yang et.al 2001). For this study, the configuration of the retrieval of the extinction coefficient is fixed for the microwindow 832.4-834.4 cm-1."

3. Somewhere (page 2, lines 13 and 14) in the introduction, it needs to be mentioned, which definition of the tropopause is used in the manuscript (in agreement with your sentence on page 12, line 26 'As discussed in Pan and Munchak (2011) different definitions of the tropopause can lead to different results.').

The selected tropopause criteria is included at the end of the introduction, where the general outline of the study is specified:

Page 3, line 6-11: "Our work analyzes the cirrus measured by GLORIA during the Wave-driven ISentropic Exchange (WISE) campaign in September/October 2017 with the purpose of obtaining more information about the nature of cirrus (both macro-physical and micro-physical properties) and thus, improve the understanding of their formation processes. The analysis includes the macro-physical properties of cirrus clouds, i.e., cloud top height (CTH), cloud bottom height (CBH), vertical extent and their position with respect to the tropopause. The tropopause was computed following the definition of the first thermal tropopause from WMO (1957)."

**4. Page 12, line 5: We rather explain it by the differences in cirrus cloud selection criteria of the studies. Isn't it also a matter of detection sensitivity for the two set of observations, and if yes, how would the inferred cloud fractions compare for the same detection threshold of the extinctions?**

The studies mentioned in page 12 regarding the frequency of cirrus clouds are Goldfarb et. al 2001 and Sassen et. al 2008. Goldfarb et. al 2001 uses ground-based lidar observations, while Sassen et. al 2008 analyses data from active remote sensors in space. The instruments of both studies have nadir viewing geometry.

The sensitivity of the instruments can play a role in the difference of the results, but in order to investigate it, a deeper research should be done, which is out of the scope of this manuscript. We argue that the main contribution to the different results comes from not using exactly the same criteria for the cirrus selection. Sassen et. al 2008 consider as cirrus only clouds with  $\tau$ <3 and with a maximum cloud top temperature of -40°C. Goldfarb et. al 2001 considers for the detection of cirrus a threshold that is defined for each nightly determination and that the cloud layer is in an air mass with a temperature of -25°C or lower. (Information added in page 13, line 14-15 and page 14, line 1-3).

Technical, grammatical and typographical corrections

**1. Throughout the manuscript, I found the arbitrary change in tenses (c.f. from simple past, to presence and vice versa) rather irritating. Check for internal consistency and the appropriateness for the used tenses.**

Thank you for pointing this inconsistency out. The manuscript has been reviewed and modified accordingly.

**2. Page 1, line 8: What is a 'cloud top bottom height'?**

This term was incorrect. It is has been changed to 'cloud bottom height'.

3. Page 2, line 30 and elsewhere: I wonder whether the notation of a ,value' is really needed to describe the magnitude of a physical quantity, c.f. This IWC value matches...., instead of .... This IWC matches; in the Figure 3 legend: CI value instead of CI; page 11, line 3: CI-values lower than  $1.2 \dots \rightarrow$  CI lower than 1.2... page 12,line 3: These values... $\rightarrow$  These fractions ..; page 16, line 24: The corresponding TPmed and TP95 have close values  $\rightarrow$  The altitude for corresponding TPmed and TP95 are close et cetera

The manuscript has been reviewed and all the non-necessary 'values' have been deleted.

**4. Page 3, line 14. .... of providing information (of what?, c.f. on cirrus clouds) in the observation gap ....**

Changed to: 'It was designed with the purpose of providing information about trace gases and temperature fluctuations in the observational gap that comprises small-scale structure...' (Page 3, line 21-23).

**5. Page 3, line 19: for measuring (what ?, c.f. microphysical parameters) of optically and vertically thin cirrus...**

Changed to: "for investigating optically and vertically thin cirrus." (Page 3, line 27).

**6. Page 3, line 22: ... horizontally averaged spectrum... Mention here the size of the horizontal dimension/footprint over which it is averaged**

Changed to: 'but the horizontally averaged spectrum (averaged over 48 pixels) of each line of the 2D array...' (Page 3, line29-30)

**7. Page 3, line 31: Table 1 summarizes the most important technical characteristics (→ features) of GLORIA.**

Changed to 'features'. (Page 4, line 10)

**8. Page 4, Table 1 legend: Observer altitude of 15 km and tangent altitude of 10 km. \*Ungermann (2020, in preparation) → (Ungermann et al., 2020, in prep.) Changed to (Ungermann, 2021, in preparation) (Page 4, Table 1 legend)**

**9.** Page 4, line 1: Provide a reference (Hoffmann, 2006) for the RT model JURASSIC2 here. Added in page 4, line 12: (Hoffman, 2008; Griessbach et. al, 2013; Ungermann et al. 2015).

**10. Page 5, line 1: define 'ice water content (IWC)' on the first occurrence in the manuscript.**

Changed to: 'ice water content (IWC), i.e, the cloud ice mass in unit volume of atmospheric air.' (Page 6, line 5)

**11**. Page 5, line 3: In addition, we retrieved the potential vorticity (PV) and equivalent latitudes...for consistency put latitude in singular, or potential vorticity into plural.**

Changed to: 'potential vorticity (PV) and equivalent latitude' (Page 6, line 8)

**12. Page 7, equation 1: Check for the correct notation**

Equation 1 has been reviewed and modified according to the notation in Griessbach et. al, (2016). (Page 8, equation 2).

**13. Page 7, line 18: ... as the percentile 95 and -86% (correct?)**

Correct. -86% refers to the percentile 5. This means that for a few cases the extinction is larger in the single scattering simulation than in the no scattering one.

14. Page 8, line 5: The range of retrievable extinction values for clouds → The range of retrievable clouds extinctions ...

Changed to: 'The range of retrievable clouds extinctions...' (Page 9, line 14)

**15. Page 8, line 20: If this gradient has a small variability, that means there are no elements that cause a sudden increase in the extinction. Please reformulated this sentence in order to make better clear what is meant.**

Changed to: 'If this gradient has a small variability, that means there are no elements, i.e aerosols or cloud particles that cause a sudden increase in the extinction and therefore a large gradient.' (Page 9, line 28-30)

16. Page 8, line 31: This value is similar  $\dots \rightarrow$  This detection limit is similar  $\dots$  Changed to: 'This detection limit is similar' (Page 10, line 5)

17. Page 8, line 34: ..... the low number of counts shifts (what counts?)...  $\rightarrow$  the low number of positive cirrus cloud detection ? ...... Changed to: 'the low number of observations and occurrences of clear sky...' (Page 10, line 8-9)

18. Page 8, line 35: This value, as well as the threshold for lower altitudes, agrees....  $\rightarrow$  Our threshold for this and lower altitudes, agrees

Changed to: 'Our threshold for this and lower altitudes agrees...' (Page 10, line 9)

**19**. **Page 9, line 6: and vertical extent.**  $\rightarrow$  **and their vertical extent** Changed to: 'and their vertical extent.' (Page 11, line 6)

20. Page 9; line 9: which the extinction (or CI) has a value equal to or larger than the kthres → which the extinction (or CI) is equal to or larger than the kthres Changed to: 'which the extinction (or CI) is equal to or larger than the kthres...' (Page 11, line 9)

21. Page 10, line 4: ....first point with an extinction (what point? and to what refers first?)
→ ....first detection in the series of limb observations with an extinction...
Changed to: 'first detection in the series of limb observations with an extinction...' (Page 11, line 13-15)

**22**. **Page 11, line 4: Thin profiles.... What are thin profiles? Profiles of small extinctions? ...** Changed to: 'Optically thin profiles, i.e. with small extinctions, are...' (Page 11, line 18, page 12, line 1)

23. Page 11, line 6: .....reaches saturation after  $CI = 1.2 \dots (after?) \rightarrow$  for CI's larger than 1.2.

Changed to: 'reaches saturation for CI's smaller than 1.2 ...' (Page 12, line 3) 24. Page 11, line 11: ...the different spectral slopes... → the different wavelength dependence

Changed to: 'the different wavelength dependence' (Page 13, line 4)

**25. Page 12, line 5: However, 60% is considerably...→ However, a fraction of 60% is considerably**

Changed to: 'However, a fraction of 60% is...' (Page 13, line 12)

**26. Page 12, line 10: ......8 and 10 km present equivalent latitudes... → ....8 and 10 km as function of equivalent latitudes**

The suggested modification would alter the meaning of the sentence. The message of this sentence is that the CTH between 8 and 10 km have mixed air mass characteristics, as some of them have equivalent latitudes of air masses in the tropics and other CTHs have equivalent latitudes of polar air masses. (Page 14, line 5-6)

**27. Page 12, line 11: For CTHs between 10km and about 12.5km the air masses have an equivalent latitude typical of mid-latitudes, whereas the highest CTHs, above about 12.5km are almost subtropical.--> CTHs between 10km and about 12.5km often occur at equivalent latitude typical for mid-latitudes, whereas the CTHs above about 12.5 km, are (were) related to subtropical latitudes.**

Changed to: "CTHs between 10km and about 12.5km often occur at equivalent latitude typical for mid-latitudes, whereas the CTHs above about 12.5 km, are (were) related to subtropical latitudes" (Page 14, line 6-8)

**28. Page 12, line 13: The main difference between both methods is the slightly higher (1 - 2 pixels) CTHs of the CI method. $\rightarrow$ The main difference between both methods is the slightly higher $(1 - 2 \text{ pixels}) \rightarrow$ CTHs inferred from the CI are slightly higher (1 - 2 pixels) than for the extinction method.**

Changed to: 'The main difference between both methods is that the CTHs inferred from the CI are slightly higher (1 - 2 grid points) than for the extinction method.' (Page 14, line 8-9)

**29. Page 12, line 14: Considering all observed profiles about 39% are optically thick using the extinction and 41% the CI method. $\rightarrow$ From all considered profiles, 39% can be characterized as optically thick (provide a number here) using the extinction method and 41% the CI method.**

Changed to: "From all considered profiles (13539), about 39% (5232 profiles) can be characterized as optically thick using the extinction method and 41% (5517 profiles) the CI method." (Page 15, line 1-2)

**30. Page 12., line 14: The maximum extinction detected for thin clouds in which a CBH was possible to determine is $4\times10-2$ km-1. $\rightarrow$ For optically thin clouds, the maximum extinction at CBH was $4\times10-2$ km-1.**

As the maximum extinction does not necessarily correspond to the CBH, we will leave the original sentence.

**31. Page 12, line 20: ...the vertical extent distribution $\rightarrow$ the frequency distribution of the vertical extent**

Changed to: 'They showed that between May and November the frequency distribution of the vertical extent of the observed clouds...' (Page 15, line 8-9)

32. Page 12, line 22: ... computed...  $\rightarrow$  ... found... Modified.

33. Page 12, line 27: ... the first thermal tropopause altitude was computed from ERA5 data .. 'first' with respect to what?

It refers to the first tropopause found when analyzing the temperature profile from the lowest altitude upwards.

34. Page 12, line 28: .... sampling air masses that can be heterogeneous. Consequently, the tropopause is usually not constant ..  $\rightarrow$  hence the sampled air masses can (could) be heterogeneous in the horizontal. Further, the tropopause height is (was) not constant. Change to: "hence the sampled air masses could be heterogeneous in the horizontal. Further, the tropopause height was not constant." (Page 16, line 2-3)

**35**. **Page 12, line 29: .... were applied... → were used** Changed to: 'Two methods were used...' (Page 16, line 3)

36. Page 12, line 33: ... the air mass at 16:18 UTC is homogeneous... → the air mass at 16:18 UTC was homogeneous (see my comment 1 above). Changed to: 'was homogenous' (Page 16, line 7)

37. Legend 7: PDFs of equivalent latitude (Eqlat) normalized for each altitude bin for (a) CTH detected with the extinction, (b) CTH detected with the CI and (c) CTH from ERA5. The altitude of the tangent points (TgPt) is the y axis  $\rightarrow$  PDFs of CTH as function of equivalent latitude (Eqlat) normalized for each altitude bin from (a) the extinction, (b) from the CI, and (c) from ERA5. The y axis shows the altitude of the tangent points (TgPt). Following also suggestions from referee 2 and 3, the caption of Figure 7 has been changed to: 'PDFs of CTH as function of equivalent latitude (Eqlat) normalized for each altitude for each altitude bin from (a) the extinction, (b) the CI and (c) ERA5, discussed in Sect. 4.3. The y axis shows the altitude of the tangent points (TgPt). The black line represents the mean tropopause height during September-October 2017 as a function of the equivalent latitude. It was computed from ECMWF analysis data."

**38**. **Page 13**, **line 1:** ... there are heterogeneous... → there were heterogeneous Changed to: 'there were...' (Page 16, line 8)

39. Page 13, line 2: .... since the CTH is above or below the tropopause depending on the chosen tropopause altitude.  $\rightarrow$  since as to whether the CTH is located above or below the tropopause depends on the chosen tropopause altitude.

Changed to: 'since as to whether the CTH is located above or below the tropopause depends on the chosen tropopause altitude.' (Page 10, line 9-10)

40. Page 13, line 11: .... both percentages decrease but still detect CBHs above the TP... → both occurrences decrease but still CBHs above the TP are detected.

Changed to: 'both occurrences decrease but still CBHs above the TP were detected.' (Page 16, line 20-21)

41. Page 13, line 12: The presence of complete layers above the tropopause is inconclusive, as these CTHs and CBHs are in general just one altitude bin apart and the CBH is only one or two altitude bins above the tropopause, which is within the uncertainties of the CBH.  $\rightarrow$  The presence of complete layers above the tropopause is inconclusive, since for the cases CTHs and CBHs only separated by one altitude bin and the CBH is only one or two altitude bins above the tropopause, which is within the uncertainties of the CBH.  $\rightarrow$  The presence of complete layers above the tropopause is inconclusive, since for the cases CTHs and CBHs only separated by one altitude bin and the CBH is only one or two altitude bins above the tropopause, which is within the uncertainties of the CBH. Changed to: 'The presence of complete layers above the tropopause is inconclusive, since these CTHs and CBHs are only separated by one altitude bin and the CBH is only one or two altitude bins above the tropopause, which is within the uncertainties of the CBH.

42. Page 14, line 5: Spang et al. (2015) analysed CRISTA data (Spang et al., 2015) and concluded with a frequency of occurrence of 5% of all observations and Zou et al. (2020) obtained 2% for CALIPSO data and 4 - 5% for MIPAS data.  $\rightarrow$  Spang et al. (2015) analysed CRISTA data for cirrus clouds and concluded to a 5% their frequency of occurrence and Zou et al. (2020) inferred their occurrence to 2% for CALIPSO data and 4 - 5% for MIPAS data.

Changed to: 'Spang et al. (2015) analyzed CRISTA data and concluded to a 5% frequency of occurrence of cirrus clouds (of all observations) and Zou et. al (2020) inferred their occurrence to 2% for CALIPSO data and 4~--~5% for MIPAS data' (Page 16, line 26-28)

**43. Page 14, line 7: .... above the tropopause derived from ERA-Interim. → above the ERAInterim thermal tropopause.**

Changed to: 'above the ERA-Interim thermal tropopause.' (Page 16, line 29)

**44. Page 14, line 9: These values are comparable to the ones of the literature. $\rightarrow$ These occurrence frequencies are comparable to those reported in the literature (provide references here).**

Changed to: 'These occurrence frequencies are comparable to those reported in the literature (Goldfarb et. al 2001, Spang et. al 2001, Zou et. al 2020).' (Page 16, line 30-31)

**45. Page 14, line 10: .... ERA-Interim, the equivalent criterion would be 0.25 km above the tropopause.--> ERA-Interim. Accordingly, an equivalent criterion would be to mandate the cirrus CTH to be located 0.25 km above the tropopause.**

Changed to: 'However, as we used ERA5 data, which has a better vertical resolution than ERA-Interim, the equivalent criterion would be to mandate the cirrus CTH to be located 0.25 km above the tropopause.' (Page 16, line 31-33)

46. Page 14, line 12: We explain the differences...  $\rightarrow$  We explain these differences... Changed to: 'these differences' (Page 16, line 34)

**47**. Legend Figure 9: The three profiles have been smoothed with a three points running mean.--> The three profiles were smoothed with a three points running mean. Changed to: 'were smoothed...'

48. Table 2, legend: Percentage with respect to all retrieved profiles of cloud top heights (CTHs) and cloud bottom heights (CBHs) detected above the median tropopause (TPmed)

and the percentile 95 of the tropopause (TP95) for both detection methods. → Percentages of cloud top heights (CTHs) and cloud bottom heights (CBHs) detected above the median tropopause (TPmed) relative to all retrieved profiles and the percentile 95 for their occurrence above the tropopause (TP95) for both detection methods. Changed to: 'Percentages of cloud top heights (CTHs) and cloud bottom heights (CBHs)

detected above the median tropopause (TPmed) and the percentile 95 of the tropopause (TP95) relative to all retrieved profiles for both detection methods.'

**49. Page 16, lines 1 -3: As explained in Sect. 2.3, one of the variables sampled following the viewing geometry of the GLORIA instrument is the IWC for ERA5, which when integrated along the LOS results in the limb IWP. $\rightarrow$ As explained in Sect. 2.3, one of the parameters sampled by GLORIA is IWP, which can be compared with the ERA5 reanalysis when the ERA 5 IWC is integrated along the LOS.**

Changed to: 'As explained in Sect. 2.3, one of the parameters from ERA5 sampled following the viewing geometry of the GLORIA instrument, is the IWC, which when integrated along the LOS results in the limb IWP.' (Page 18, line 4-6)

**50. Page 16, line 5: ... this is caused by the fact that for large ...> this is since for large particles...**

Change to: 'this is since for...' (Page 18, line 7)

**51. Page 16, line 9: Figure 7c shows a similar distribution of CTHs in ERA5 data as the one derived from the measurements ... --> Figure 7c shows a similar pattern of CTHs inferred from ERA5 data as those derived from the measurements.**

Changed to: 'Figure 7c shows a similar pattern of CTHs inferred from ERA5 data as those derived from the measurements.' (Page 18, line 11)

52. Page 16, lines 9 – 11: The fraction of CTHs detected in ERA5 is about 59% of all profiles, the same as the one of the CI method (59%) and only slightly lower than the fraction for the extinction method (61%).  $\rightarrow$  From all investigated profiles, the fraction of detected CTHs is 59% from ERA 5, 59% using the CI method, and 61% extinction method. Changed to: 'From all investigated profiles, the fraction of detected CTHs is 59% from ERA5, 59% using the CI method and 61% the extinction method.' (Page 18, line 12-13)

**53. Page 16, line 12: .... to not considering ... which would mean increase the number of CTHs observed between 8 and 11 km $\rightarrow$ discarding.... increases the number of CTHs incorrectly attributed to 8 and 11 km altitude range.**

Changed to: 'This could be related to discarded multi-layer clouds in the detection algorithm, which could increase the number of CTHs observed between 8 and 11 km as also CTHs below the first CTH of the multi-layer cloud would be included.' (Page 18, line 14-15)

**54**. **Page 16, line 17: .... than for ERA5.**  $\rightarrow$  **than reanalysed in ERA5.** Changed to: 'more cirrus measured by GLORIA than present in ERA5' (Page 18, line 19)

55. Page 16, line 17: Considering all occurrences above the TPmed, the observations detect about 50% more than ERA5 data-set. → When considering all occurrences of cirrus above

**the TPmed, the observations indicate 50% more cirrus clouds than found in ERA5.**

Changed to: 'When considering all occurrences of cirrus above the TPmed, the observations indicate 50% more cirrus clouds than found in ERA5.' (Page 18, line 19, page 16, line 1)

**56. Page 16, line 21: In Sect. 4.2 the presence of complete layers above the tropopause was suggested,... $\rightarrow$ The analysis presented in Sect. 4.2 suggest the presence of complete cirrus layers located above the tropopause.**

Changed to: 'The analysis presented in Sect. 4.2 suggest the presence of complete cirrus layers located above the tropopause.' (Page 19, line 4)

**57. Page 16, line 23: .... Only cloudy points... (what are cloudy points?) -> For both the extinction method and CI, measurements with a positive cloud detected are marked by colours.**

Changed to: 'For both the extinction method and CI, measurements with a cloud detection are marked by colors.' (Page 19, line 6-7)

58. Page 17, lines 1 - 3: The CBH is slightly higher for the extinction method and above the tropopause, but still within the detection error, therefore, no affirmation of it being undoubtedly above the tropopause is made.  $\rightarrow$  For the extinction method, the CBH is located slightly higher than for the Ci method but still within the detection error. Therefore, the cirrus can't unambiguously be ascribed to locations above the tropopause.

Changed to: 'For the extinction method, the CBH is located slightly higher than for the CI method but still within the detection error. Therefore, the cirrus cannot unambiguously be ascribed to locations above the tropopause.' (Page 19, line 10-12)

**59. Page 17, line 3: In the location... $\rightarrow$ At the location**

Changed to: 'At...' (Page 19, line 12)

**60. Page 17, line 7: These values of PV and N2 indicate... → Therefore both the PV and N2 indicate...**

Changed to: 'Therefore both the PV and N2 indicate...' (Page 19, line 18)

**61. Legend Figure 11: Describe the grey line (i.e. the flight trajectory).**

Changed to: '...Black triangles indicate the CTHs and the white circles the CBHs. The grey line marks the flight trajectory...'

**62**. **Page 18, line 2: ... and the derived extinction ... → and the inferred extinction...** Changed to: 'retrieved extinction...' (Page 19, line 24)

**63. Page 18, line 3: ... and did not include...→ and excluded**

Changed to: 'and excluded multi-layer clouds.' (Page 19, line 25)

**64. Page 18, line 5: .... with an extinction of $2 \times 10-4$ km $-1.... \rightarrow$ with an extinction as low as $2 \times 10-4$ km-1.**

Changed to: 'extinction as low as...' (Page 19, line 26)

**Observation of Cirrus Clouds with GLORIA during the WISE Campaign: Detection Methods and Cirrus Characterization**

Irene Bartolome Garcia1, Reinhold Spang1, Jörn Ungermann1, Sabine Griessbach2, Martina Krämer1, Michael Höpfner3, and Martin Riese1

[revised manuscript text omitted]

**2.3 Meteorological dataset**

\_

We used the high resolution ERA5 data-set dataset provided by the European Centre for Medium-Range Weather Forecasts (ECMWF). The reanalysis data are available at 31 km horizontal resolution at 137 levels from surface to 80 km (Hersbach et al., 2020). The ERA5 dataset provides hourly data for a large variety of meteorological and climate variables. To perform the

comparison between model and measurements, the variables of interest were sampled according to the GLORIA measuring geometry, as shown in Fig. 2. This figure represents the limb geometry during one measurement. For every line-of-sight (LOS)LOS, every 30 km, the meteorological variables were computed from the corresponding parameters of the ERA5 data set, i.e. first thermal tropopause (TP), equivalent latitude and ice water content (IWC)-, i.e, the cloud ice mass in unit volume of atmospheric air. As the signal is integrated along the LOS of the instrument, the same applies for the IWC, thus the final parameter used for the comparison is was the limb ice water path (IWP), i.e. the IWC integrated along the LOS (Spang et al., 2015). In addition, we retrieved the potential vorticity (PV) and equivalent latitudes latitude from the ECMWF data at the tangent point. The static stability (N2) used to analyze the stability of the atmosphere was computed from GLORIA retrievals. The potential temperature static stability is the square of the Brunt-Vaisala frequency (N), defined as:

$$N = \sqrt{\frac{g}{\theta} \frac{\partial \theta}{\partial z}},\tag{1}$$

15 where g is the local acceleration of gravity,  $\theta$  is the potential temperature, and z is the altitude of the air parcel. N2 describes the vertical temperature stratification of the atmosphere and gives an insight of if an air parcel is in a transition region between

---

## Author Comment (AC2) · 29 Jan 2021

**Final response to referee #2 comments on paper amt-2020-394**

First of all, we would like to thank anonymous reviewer #2 for the constructive comments, which helped to improve the manuscript.

The comments are in bold letters and the answer in normal format. The page and line indicated in the answers correspond to the new version of the manuscript. At the end of the comments the revised manuscript is attached, with the additions in blue and the deletions in red.

**Anonymous referee #2**

Specific comments

**1. L. 8, p.1: "cloud top bottom height": Do the authors mean: "cloud bottom height"? Otherwise, what is the difference between "cloud top bottom height" and "vertical extent"?**
The authors mean cloud bottom height. It has been corrected in the manuscript. (Page 1, line 9)

**2. L. 9, p.1: What do the authors mean by "fraction of cirrus clouds"? Is it an estimate of covered area referred to the global coverage? Is it restricted to some altitude range with respect to the tropopause level? Is there some time reference in terms of annual mean? The estimate of 13 to 27 % is different from what is mentioned in L. 18, p.1: is it a number derived from the GLORIA measurements?**
By 'fraction of cirrus clouds' we mean the percentage of the total number of observations made with GLORIA in which a cloud has been detected. Even the altitude range of the observations is from 6 – 15 km, the analysis is restricted to 8 – 15 km to minimize the influence of the water vapor continuum at lower altitudes. The estimated of 13 – 27% is referred to the duration and area of the WISE campaign and derived from the GLORIA measurements, depending on the detection method and the definition of the tropopause. Therefore, it is different to the one in L. 17, p.1, which refers to a global average during 1 year.

**3. L. 12, p.1: What do the authors mean by "unattached cirrus layers"?**
By 'unattached cirrus layers' we mean cirrus that are not associated with any other cloud.

**4. *L. 17, p.1: Do the authors mean "one or banks of small white flakes"?***
'one' refers to the second genera of high clouds mentioned in L.17, p.1. Therefore, the intention of this sentence is to explain that the second genera (cirrocumulus) consist of banks of small, white flakes. However, following the suggestion of referee #3, the sentences 'It is possible… cloud veils' (Page 1, line 15-17 original manuscript) have been deleted.

**5. L. 20, p.1: "due to the low temperature of their environment": I guess the scattering due to the presence of the cirrus is the main driver, and not the temperature: I guess an even cold, dry, cloud-free atmosphere is not equally absorbing than cirrus clouds in the same temperature conditions. Please clarify.**

This sentence has been modified to: "Cirrus clouds generally have a strong infrared greenhouse effect, leading to warming. However, optically thick cirrus with many ice crystals of a few micrometers, can have the opposite effect (Kraemer et. al 2016) " (Page 1, line 19-20)

**6. L. 21, p.1: "they influence the amount of solar radiative energy received": Is this statement not in contradiction with L. 19-20, p.1: "cirrus clouds are rather transparent to incoming solar radiation"?**
We do not consider it is a contradiction, as the influence of clouds in the radiative energy received can be due to absorbing it, or letting the light passing through them.

**7. L. 21-22, p.2: "2% of stratospheric cirrus : : : 4-5% for MIPAS in middle latitudes": To which quantity do these percentages refer ? To the global cloud occurrence frequency? On an annual basis?**
These percentages refer to a six year mean global distribution (2006 -2012). Information added in page 2, line 21-22.

**8. L. 35, p.2: What do the authors mean by "long light of sight"?**
As shown in figure 2, the line of sights extends several hundred km.
By 'long line-of-sight' we want to indicate that the path of the ray through the tangent layer is quite long (in the order of 226 km for a tangent altitude of 10 km).

**9. L. 2, p.3: What do the authors mean by "nature of the cirrus"? Do they mean the optically thin/thick or subvisible character, or the (microscopic) structure of the clouds, or something else?**
We mean any characteristic of the cirrus, both macrophysical properties (cloud top height, cloud top bottom, vertical extent) and microphysical (IWC, radius). Information added in page 3, line 7-8.

**10. L. 14-20, p.3: There are some repetition of information provided in p.2. The authors might consider remove them.**
Even there are some technical features of GLORIA that are repeated in the introduction and then in the section about the instrument, we prefer to leave it like it is now to emphasize why using GLORIA is interesting.

**11. L. 23, p.3: Except saving memory space, does such averaging present any advantage in better visualizing the vertical structure, by getting rid of the local variability? Table 1, p.4: This table seems to provide properties of both detectors (Fourier-transform spectrometer and 2D detector). It might be important to specify which one is concerned by each property. For instance, are both instruments covering the same range 780- 1400 cm-1?**
No, the horizontal averaging does not improve visualizing the vertical structure. It improves the signal to noise ratio. The GLORIA instrument is formed by a Michelson interferometer and a 2D detector array that registers the beam coming out of the interferometer. The 2D detector produces interferograms that are transformed to the space domain to produce calibrated spectra using the Fourier transform. Table 1 provides information about the GLORIA instrument as a whole.

**12. Figure 1: The choice of map does not render very well the location of land, and specifically**
**of countries of importance, like Ireland and UK. Has the colour code (e.g. with all range of blue colours at the level of the sea and oceans) any importance for the present study? If yes, the meaning of the colour code and a colour map should be provided. If not, I suggest to change the choice of map to make the information of importance (I guess: the geolocation) clearer and more visible.**
The colors of the map are of no importance for this study, therefore, the map has been changed to a white background, with the country lines in black and the land in grey.

**13. L. 4-5, p.5: It seems useful to give some more insight into what static stability is, by giving some value of the static stability for extreme cases and/or by giving some equations that would also be useful to explicit the temperature dependence mentioned in l. 6, p.5. The quantity N should be defined. This paragraph indicates that the static stability is an important parameter, but it is not clear why, and for which purpose the stability of the atmosphere has to be computed.**
The following explanation has been added (page 6, line 8-15):
'The static stability ($N^2$) used to analyzed the stability of the atmosphere was computed from GLORIA retrievals. The static stability is the square of the Brunt-Vaisala frequency (N), defined as:

$$N = \sqrt{\frac{g}{\theta}\frac{\partial \theta}{\partial z}}, \qquad (1)$$

where g is the local acceleration of gravity, $\theta$ is the potential temperature, and z is the altitude of the air parcel. $N^2$ describes the vertical temperature stratification of the atmosphere and gives an insight of if an air parcel is in a transition region between the troposphere, characterized by low $N^2$ ($N^2 \approx 1 \times 10^{-4}\ s^{-2}$) and the stratosphere, characterized by high $N^2$ ($N \approx 5 \times 10^{-4}\ s^{-2}$) (Grise et. al 2010).'

**14. L. 10, p.5: If possible, I suggest to add some reference in English providing a description of JURASSIC2 for the more general readership of the journal.**
The following references have been added (page 6, line 21-22):
(Hoffman et al., 2008; Griessbach et al., 2013; Ungermann et al., 2015)

**15. L.10, p.5 - l.5, p.6: In view of the very scarce documentation provide about JURASSIC, I suggest to extend a little bit the description, and make it somewhat less cryptic. E.g., concerning the retrieval technique, what is the retrieved quantity, and from which quantity-ies? If the inversion technique has some similarities with the one used for limb sounding measurements or any other techniques, this might be mentioned. A reference is needed about the Schwarzschild equation.**
The retrieved quantity in this study is the extinction coefficient (page 6, line 30). From Sec. 2.4 (where JURASSIC2 is introduced) the reader is referred to Sec. 3.2 for an explanation about the extinction retrieval. To retrieve the extinction, an inverse problem must be solved, where the measurements are the radiances in the spectral region 832.4 – 834.4 cm$^{-1.}$ A reference to Ungermann et al. (2015) with a detailed description of the retrieval is added. (Page 9, line 2).

JURASSIC2 is adequate for the retrieval of trace gases, temperature and the extinction coefficient from measurements of both remote sensing limb and nadir sounders in the infrared (Griessbach et. al, 2013). This feature is now specified in page 6, line 22: 'It combines a forward model with retrieval techniques (for both limb and nadir geometries) and allows us…'

A reference about the Schwarzschild equation is added in page 6, line 24: (Petty, 2006; Wallace and Hobbs, 2006).

**16. L. 11-12, p.6: What do the authors mean? Cloud index and extinction coefficient are physical quantities, and not methods.**
We mean two different ways of identifying clouds in our measurements. The first way is using the CI and the second way the extinction coefficient.
The sentence has been changed so it is clearer: 'To analyze the data, two methods to identify optically and vertically thin clouds at high altitudes were used. One method used the cloud index and the other the extinction coefficient.' (Page 7, line 1-2).

**17. L. 16-17, p.7: "The difference (: : :) is 21%": please specify the metric used.**
The metric used is the mean of the median of both flights. Added in page 8, line 18: 'The difference (calculated as the mean of the median difference of both flights) between the extinction neglecting…'

**18. L. 12-19, p.7: From this paragraph, it appears that the present case is a rather unfavourable case with respect to the study by Höpfner and Emde (2005). From runs on test cases that are not described, a quite large difference is found between the retrieved extinction coefficient using no-scattering approximation, and the retrieved extinction coefficient using single scattering (which is a simplified case with respect to the actual multiple scattering case). In the no-scattering case, quite high values are found for the most extreme cases (P5 and P95). From all these data, it is difficult to convince (at least the non-expert) that a non-scattering approach is sufficient. The authors might provide the values of the difference at 2 x the standard deviation (P16 and P84) or interquartile values or another useful metric. They also should specify how the test cases were chosen (using different typical cirrus cloud configurations? Was the distribution of the different kinds of situations representative for real atmospheric conditions at the considered latitude?). Finally, an important and more convincing argument would be to provide an estimate of the uncertainty on all target macrophysical quantities and on the detected cirrus cloud fraction, possibly in function of the latitude.**
We have added the values of the difference at P16 and P84 in page 8, line 20-21. The mean difference at $2\sigma$, i.e. percentile 16 and percentile 84 is -4% and 49%, respectively.

The two flight selected for the single scattering run were chosen because both thin cirrus and thick cirrus were observed, and therefore, constitute an interesting case for studying the influence of scattering in different cases. Added in page 8, line 17-18.

For the flights with single scattering, the CTH was computed following the same procedure as for the flights with no scattering. The agreement in the detection of a CTH was 98% for both study cases. The altitude of the CTH is for ~ 71% of the profiles the same, being the typical difference 0-0.375 m. Following the same procedure for the CBH, the agreement in the determination of a CBH for the detected CTH is 90%, obtaining the same altitude in 58% of the coincidental profiles. The typical difference for the CBH is also 0 – 0.375 m. The detected clouds for both flights and both runs cover the range 45°N - 75°N, with the largest occurrence between 55°N – 75°N. This information has been added in page 8, line 21-27.

**19. L. 27, p.7: What do the authors mean by "elevation angle offset"? A reference to a paper about the instrument might be useful.**
The instrument has different elevation angles that correspond to different tangent altitudes of the lines of sight. The elevation angle offset is the error in the determination of this angle.
A detailed explanation of the instrument and its concept is given in Friedl-Vallon et al. (2014) and Riese et al. (2014) (Page 3, line 18)
The reference for the characterization of the errors related to the instrument and the retrieval is Ungermann et. al 2021 in preparation (Page 9, line 12)

**20. L. 1-2, p.8: Should this error of 125 m be added to GLORIA's 140 m-vertical resolution? How does it affect GLORIA's ability to detected small-scale structures mentioned in §2.1?**
Both quantities should not be added. 125 m corresponds to the error in the model grid, whereas 140 m corresponds to the vertical sampling at a certain tangent altitude and observer altitude. The model includes corrections of the field of view and the sampling. In Sect. 2.1 we mention structures of about 500m, which can be detected by GLORIA.

**21. Caption Figure 3: "First filtering of optically thicker regions": Do the authors refer to the aerosol contributions? It would be useful to specify.**
Aerosol contribution are usually much weaker than cloud emissions. When the clouds are optically thick, GLORIA is not able to measure anything beyond them and the radiance profiles become saturated. By "first filtering of optically thicker regions" we refer to the layers where the clouds are too thick and can affect the tangent layers above.  These information has been added in the caption of Figure 3 (page 10).

**22. L. 12-13, p.8: "Above the clouds": please specify the vertical range affected by this regularization effect. Which method is supposed to be affected by this problem?**
The vertical range affected by this effect is between 1-2 grid points (0.125 – 0.250 km). This is a general effect of the extinction retrieval. Added in page 9, line 20.

**23. L. 18-19, p.8: A reference is needed for this choice of criteria for clear sky conditions. "CI always greater than 2": What do the authors mean by "always"?**
This criteria was selected specific for this study. After the selection of clear sky profiles, cross-sections for all flights (similar to Figure 3), were plotted just with the selected profiles to ensure that no clouds where being included.
By always we mean that the single profiles have a CI greater than 2 for all altitudes.

**24. L. 18-20, p.8: In which extend is this pre-selection coarse? Does it miss cloudy conditions, does it identify too much clear-sky conditions as cloudy ones, or randomly fails to distinguish clear-sky and cloudy conditions?**

To check the performance of both algorithms, we plotted cross-sections of cloud index and extinction for all flights only with the clear sky profiles to compare both selection criteria (similar to Figure 3). The pre-selection identifies less clear sky conditions than the vertical gradient of extinction criteria and at the same time identifies as clear sky profiles that contain clouds. These falsely identified profiles are not included in the second selection, confirming that the second selection yields more reliable results.

**25. Caption Figure 4: "Clear sky profiles of the vertical gradient (: : :)" looks strange. Wouldn't it be better to write "vertical gradient of the extinction coefficient in the case of clear sky conditions" ?**

Changed to the suggested sentence. (Page 11)

**26. Figure 4: I guess the reason why the width of the scattering cloud is quite constant down to ¯8 km, and then rapidly increases with decreasing altitudes (linked afterward with the water vapour influence), is related to the presence of the tropopause level. The authors could usefully add some line representative for the tropopause level for the sake of clarity.**

Figure 4 has been changed to the following figure. It shows the median (10.1km) and mean tropopause (10.7 km) of all clear sky profiles.

[Figure]

**27. L. 23-24, p.8: Do the authors expect to miss many cirrus events by using this apparently very conservative criteria (following Fig. 4)?**

Figure 4 contains the threshold used for the selection of clear sky conditions (please, read answer to comment 24).

Giving an exact value about the false positive or false negative would require a study with synthetic data where we know where the clouds are. In general, we do not expect neither miss nor identify too many. The described threshold (Figure 5) has been created to identify clouds,

especially optically thin ones and to prevent the false identification of clouds as aerosols. So far, avoiding this confusion has been achieved. Using the methods from Griessbach et al. (2014; 2016) to discriminate between ice and volcanic ash and ice and aerosols such as sulfuric acid, has yield positive results, as almost all grid points identified as clouds where correctly identified.

**28. L. 28, p.8: PDF should be defined.**
Added in page 10, line 2.

**29. L. 30, p.8: Where is the value 2 x 10-4 km-1 coming from? Figure 4 provides a value of _1.1 x 10-4 km.**
This value is coming from Fig.5, right plot, where the threshold for extinction is plotted as the black line. This plot is tangent altitude vs extinction, whereas Fig.4 is tangent altitude vs gradient of extinction. Therefore, Fig.4 cannot be used to infer the value of 2 x 10-4 km-1. To avoid further misunderstandings, more information has been added:
'This threshold is sensitive to structures with very low extinction, down to 2 x10-4 km-1 for a tangent point between 11.5km and 15km (Fig.5).' (Page 10, line 4-5).

**30. L. 31-32, p.8: It would be useful that the authors provide here the estimates by Sembhi et al. (2012) and by Griessbach et al. (2020).**
As Griessbach et al. (2020) provides an overview for several instruments, a reference to Table 1 of the cited study is added.
'This detection limit is similar to the one provided by Sembhi et al. (2012) for MIPAS, with an extinction detection limit above 13 km of 10-4 km-1 and to the findings of Griessbach et al. (2020) specified in Table 1 of the cited study.' (Page 10, line5-7).

**31. L. 32-34, p.8: Where are the estimates of the CI threshold and considerations about "low number of count shifts" coming from ?**

For each altitude bin we counted the number of cases, realizing that the number of occurrences decreases with higher altitudes, as well as the overall number of observations. Therefore, we consider that there is not enough data, and therefore, the shift of the CI threshold towards high values is unrealistic. Moreover, considering previous literature (Sembhi et al. 2012, Spang et al. 2012) we considered correct to stablish a constant CI of 5 for altitudes higher than 12 km.

**32. L. 34, p.8-L.1, p.9: Similar to comment on L. 31-32, p.8.**
The thresholds are altitude dependent, therefore in this case is not possible to provide just one value, as we want to indicate that our threshold in all its altitude range is similar to the ones in the cited literature.

**33. L.3, p.12: It would be interesting to mention the percentage of profiles showing cloud occurrence detected by both methods, to see in which extend these methods have similar performance, or can be considered as complementary. This information would usefully complete the distribution shown in Figure 7.**
The extinction method detects a total of 8247 profiles with a CTH, whereas the CI 7964. Both method present a CTH for the same 7913 profiles, i.e, for ~58% of the total of the profiles. Information added: '58% of all profiles show a CTH for both methods, which indicates a similar performance.' (Page 13, line 9-10).

**34. L. 10-11, p.12: Is there any possible confusion between the occurrences of cirrus clouds and of polar stratospheric clouds ?**
As the polar stratospheric clouds appear during winter, even if some of the flights reach polar latitudes, we do not consider it could be possible because our measurements were taken at the end of summer, beginning of autumn.

**35. L. 14, p.12: Same as comment on L.3, p.12.**
The information about how many profiles are optically thick for both methods simultaneously has been added in page 15, line 2: "36% of all profiles are optically thick for both methods".

**36. Caption Figure 7: The authors might consider adding, for Fig. 7c, that this plot is discussed in Section 4.3.**
Following this suggestion and also a suggestion from referee 3, Fig.7 has been changed, and also its caption.
"PDFs of CTH as function of equivalent latitude (Eqlat) normalized for each altitude bin from (a) the extinction, (b) the CI and (c) ERA5, discussed in Sect. 4.3. The y axis shows the altitude of the tangent points (TgPt). The black line represents the mean tropopause height during September-October 2017 as a function of the equivalent latitude. It was computed from ECMWF analysis data."

**37. Figure 8: If the distribution shows the number of estimated CTH, I guess it is the same distribution as the distribution of detected cloud occurrences. The integral of this histogram should be about 100%, since all not considered values (values < 0 or > 6 km) are unrealistic, thus supposed to be at worst marginal. However, a first estimate gives a total of ~36% in the case of the extinction method and even significantly less in the case of the CI method. What is wrong? The estimates in L. 18-19, p.12 (31% [20%] of the clouds detected by the extinction [CI] method) look consistent with Figure 8.**
Figure 8 is the distribution of the vertical extent of the observed clouds. Only for optically thin clouds it is possible to define a CBH, and therefore their vertical extent. Thus, the percentage does not add to 100%, as this is not the case for all detected clouds. The percentage given in Fig.8 is over the total number of CTHs (i.e. both optically thick and thin).

**38. Figures in supplement, L. 5, p.4: The supplement includes 15 figures seemly aimed at providing the data for each individual flight. However, flight 1 is missing, and flight 3, already illustrated in Figure 3, is duplicated. Is it what the authors want? Also, the last figure, i.e. Figure S15, corresponds to the 16th flight although a total number of 15 flight is mentioned in L.5, p.4. This should be corrected for the coherence.**
The WISE campaign consist of a total of 16 flights. The first one was a short test flight, with no scientific purpose. Therefore, there is 1 test flight and 15 scientific flights. The analyzed flights are running from the 2$^{nd}$ to the 16$^{th}$. To avoid further misunderstandings, this information has been added:
'With a total of fifteen scientific flights (plus a first test flight) (Fig.1)…' (Page 4, line 17-18)
In the supplement we would like to include all 15 scientific flights (F02-F16) to have an overview of all of them.

**37. Caption Figure 10: "all cloud top heights (CTHs) for the extinction method", for 10a and 10b; "with color code as equivalent latitude".**
Following the suggestion of referee 3 regarding the color code of Fig. 10, the caption has been modified: "Distribution for (a) all cloud top heights (CTHs) for the extinction method and (b) CTHs for extinction method above the median tropopause ($TP_{med}$). Colors indicate the tangent point altitude (TgPt)."

Technical corrections
**L. 2, p.2: "their detection"**
Modified (now page 2, line 1).

**L. 21-22, p.2: "4-5% for MIPAS at middle latitudes"**
Modified (now page 2, line 21).

**L. 4, p.4: "the data (: : :) were: : :"**
Modified (now page 4, line 16)

[revised manuscript text omitted]

---

## Author Comment (AC3) · 29 Jan 2021

**Final response to referee #3 comments on paper amt-2020-394**

First of all, we would like to thank anonymous reviewer #3 for the very helpful and constructive comments, which served to improve the manuscript. We hope that after addressing all comments the manuscript is easier to understand and the characteristic of the instrument and the viewing geometry, as well as the results are clearer and better explained.

The comments are in bold letters and the answers in normal format. The page and line indicated in the answers correspond to the new version of the manuscript. At the end of the comments the revised manuscript is attached, with the additions in blue and the deletions in red.

**Anonymous referee #3**

**Major comments**
**My main concern with the paper is that while I'm familiar with many remote sensing and in-situ detection techniques, and with the cloud climatologies they enabled, I had trouble interpreting the results presented in this paper and putting them in perspective against other retrievals. I can't believe I will be the only one. In the minor comments below I try to explain where I had the most trouble understanding.**
**I am also concerned about the uncertainty in the horizontal location of retrieved clouds associated with the retrieval technique, and I would like to see it discussed, especially when relating the altitudes of detected clouds with tropopause.**
The long line of sight of the limb viewing instruments is at the same time its advantage and weak point. An advantage because, as the signal is integrated along a long path, it is sensitive to optically thin clouds, which might be invisible to other instruments. However, this means losing information about the exact position of the clouds. In the following, we try to address all comments to highlight the usefulness of this technique concerning optically thin clouds and try to clarify all doubts.

**Minor comments**
**1. p.1, L. 15-17: "It is possible... veils" this is generic information, which has no bearing on the following discussion and actually creates confusion, as the reader might believe it needs to retain the difference between cirrus, cirrocumulus and cirrostratus to understand what follows. I think it can be omitted.**
The indicated lines are omitted in the new manuscript.

**2. P.2, L.2: "its" => "their"**
Modified (now page 2, line 1)

**3. P.2, L.2: I don't understand what are the "typical operational weather satellites" that are equipped with nadir sounders? Please be more specific. To my knowledge "typical operational weather satellites" are geostationary and provide visible/infrared imagery with near-global coverage. "Nadir sounders" make me think of active sensors, such as radars and lidars, which are not found aboard typical operational weather satellites.**

The sentence has been changed to: "which complicates its detection by active as well as passive nadir sounders." (Page 2, line 1).

**4. P.2, L.15: "Cloud and Aerosol Lidar" – this is not the meaning of the CALIOP acronym. Please fix.**

Modified to Cloud-Aerosol Lidar with Ortogonal Polarization (now page 2, line 13)

**5. P.2, L.15: "as Dessler (2009)" why mention this?**

We mention Dessler (2009) because we want to emphasize the importance of the definition of the tropopause, as the same data set with different definitions yield different results regarding stratospheric cirrus. Both, Pan and Munchak (2011) and Dessler (2009) use the same data set, but obtain different results.

We have rephrase the sentence, so its meaning is clearer: "While Dessler (2009) indicated the existence of a substantial amount of cirrus clouds above the tropopause Pan & Munchak (2011), using the same CALIOP data, demonstrated that the amount of cirrus clouds above the tropopause strongly depends on the definition of the tropopause. Moreover Pan & Munchak (2011) concluded that there is not enough evidence of clouds above the tropopause in mid-latitudes."(Page 2, line 12-15).

**6. P. 2, L.16: "there is not enough evidence of clouds above the tropopause in midlatitudes" not enough evidence for what? Please be aware that more recent papers look at cirrus clouds above tropopause using CALIPSO data: e.g. Dauhut et al. 2019 https://doi.org/10.5194/acp-20-3921-2020**

By "enough evidence" we mean that there are studies about the existence of stratospheric cirrus, with focus in mid-latitudes, but the frequency of occurrence varies and there is not a wide agreement. Here we summarize the state-of-knowledge from previous studies. We rephrased to make it clearer (please see answer to comment 5). However, in the introduction we refrain from referring to Dauhut et al. 2019 as their inferred diurnal cycle is to a large extent a measurement artefact due to different sensitivities at day and nighttime, see e.g. Zou et al. (2020).

**7. P. 2, L.22: "LIDARs" => "lidars"**

Modified (now page 2, line 22)

**8. P. 2, L.24: If you are interested in SVC cover derived from CALIPSO, please be aware of Martins et al. 2010 https://doi.org/10.1029/2010JD014519, Reverdy et al. 2012 https://doi.org/10.5194/acp-12-12081-2012, or Wang et al. 2019 https://doi.org/10.1029/2018JD029845**

We appreciate the recommendation of these very interesting studies. We have included them in the introduction:

"Martins et al. (2010) analyzed CALIOP measurements over 2.5 years and gave an insight into the global frequency of SVC, being more common in the tropics (30-40%). Reverdy et al. (2012) reported a significant population of SVC in the tropical upper troposphere. However, Davis et al. (2010) found that CALIPSO would be missing about 2/3 of SVCs with $\tau < 0.01$. Due to the long path of the line-of-sight (LOS) through the cirrus, typical of limb instruments, clouds that might be invisible to the nadir viewing instruments, are detectable." (Page 2, line 26-30)

**9. P. 2, L.26: Again, which nadir viewing instruments?**
We mean nadir viewing instruments in general. Our intention is to emphasize that due to the long line of sight of the limb viewing instruments, optically thin clouds can be measured by limb instruments, but might be invisible to the nadir ones.

**10. P. 2, L.33: I understand that this is the resolution of a pixel looked on straight from the plane, i.e. in a plane normal to the direction of view. What is the volume of air described by the measurement in a straight line away from the plane, I.e. along the line of sight? Is the tangent altitude the point of lowest altitude along the line of sight? Please mention these precisions.**
Due to the nature of the limb retrieval technique, specifying the volume of air measured by the instrument is a challenging task. The lines of sight along which the measurements are taken, are infinite, therefore, the volume would also be infinite. Even restricting it to the atmosphere is not necessarily sensible as the amount of emitting molecules decreases exponentially with increasing altitude. Practically, we restrict it for this computation to the altitude of the aircraft. The point spread function (PSF) of the instrument also doesn't have a compact support such that the horizontal/vertical extent of the measured cone needs to be defined in some fashion to be finite. The tangent point is defined as the point closest to the earth of the idealized pencil-beam thin line of sight. The PSF of a pixel can be computed astonishingly well from a theoretical Airy disk using the aperture of the instrument of 3.6cm and using here 830cm$^{-1}$ as a reference wavelength. The vertical width would then correspond to 0.032° and horizontally we observe 1.53°. This corresponds to a rectangle of 15m$^2$ area one km away from the aircraft, growing quadratically with distance. For a line of sight pointing to a tangent point (neglecting refraction for simplicity) 3km below the aircraft, the tangent point would be roughly 200km away corresponding to an observed area of roughly 600000 m$^2$ at this point. As such the volume of air covered by this pyramid between aircraft and tangent point is roughly 40km$^3$. Following it further until it reached again the altitude of the aircraft gives an observed volume of 319km$^3$.

The definition of the tangent altitude being the point of lowest altitude along the line of sight has been added in the caption of Figure 2 (following comment 13) and in the first appearance of the term tangent altitude: '…at a tangent point altitude (i.e. the closest point of the line of sight to the Earth's surface) of 10 km…' (Page 3, line 1).

**11. P. 3, L. 9: "step stone" => "stepping stone"?**
Changed to: 'stepping stone' (Now page 3, line 16).

**12. P. 3, L. 23: "one profile": what is the vertical resolution of one such profile?**
The vertical resolution is given by the full width at half maximum of the averaging kernel, which describes the relation between the retrieved quantities and the true atmospheric state.

The following figure illustrates the vertical sampling of one profile for an altitude observer of 14.21 km for all tangent point altitudes. The right plot indicates the altitude of the tangent points and the left plot the corresponding pixel row of the detector. Each pixel has the same point spread function (PSF) described by an Airy disk with an aperture of the instrument of 3.6cm and using 830cm$^{-1}$ as a reference wavelength. The vertical sampling is higher, the closest the tangent point altitude is to the observer altitude, as the projection of the PSF gets wider the further the tangent point is. We have added in the manuscript an example: "The vertical sampling is higher the closest the tangent point altitude is to the observer altitude, as the projection of the PSF gets wider the further the tangent point is. For example, if the observer altitude is 14.7km, at a tangent point of 13km, the vertical sampling is about 88m, at 10km of about 150m and at 8km of about 179m." (Page 4, line 2-3).

[Figure]

**13. P. 6, Figure 2: I'm not overly familiar with measurements like GLORIA's, so please forgive me if I ask obvious questions. Here are a few things I think I've understood, that I don't think are explicitly stated before the figure, and that would help its understanding if they were:**
**- I understand each blue line in Figure 2 to be a different line of sight.**
Correct.
**- I understand GLORIA scans successive lines of sight, each corresponding to a different angle with the horizontal, and each describing cloud information along the line of sight that corresponds to that angle and to a minimum altitude level.**
Correct.
**- I understand that the cloud information (IWC) is integrated along the line of sight, and the obtained IWP used as cloud indicator. It follows that the instrument documents the vertical distribution of clouds across a very large horizontal distance that varies with altitude, eg for 7km ASL the horizontal distance described is 420km, for 15km it is 660km, for 18km it is 720km, etc (according to the lines of Figure 2).**
For determine the presence of clouds with GLORIA we do not use IWP. IWP is used only for the ERA5 data, as a way to identify clouds in the ERA5 data set, adapted to the viewing geometry of GLORIA. This way, we are be able to compare the ERA5 data set with the GLORIA results. GLORIA measures radiance and this signal is the one that is integrated along the line of sight.

The cloud index, i.e the ratio of the mean radiances between the spectral region 788.2 – 796.2 cm$^{-1}$ and the spectral region 832.4-834.4 cm$^{-1}$, already used in other studies, is one of the methods we use to identify clouds. The second method is the extinction coefficient, which is retrieved using the model JURASSIC2 to solve the inverse problem in which the radiance is the input.

Related to the horizontal distance, as you can see in the following figure, the line of sight would go twice through an altitude of 7 km, first at a horizontal distance of 190 km and then of 440 km. 15 km would be at a horizontal distance of about 680 km and 18 km at around 730 km (yellow rectangles). The lowest line of sight is the longest one.

[Figure]

For other line of sight, with a tangent altitude point, e.g., at 12 km, the horizontal distances would change. This line of sight would not go down to 7km, 15 km would be at 450 km and 18 km at 520 km.

[Figure]

**- I understand that analysis of GLORIA data can tell a cloud is located along a given line of sight, which provides some insights into its spatial location. Each value of GLORIA's vertical cloud profiles is representative of a quite long path, that is longest at highest altitudes, and neither horizontal nor vertical: for instance, the lowest line of sight in Figure 2 will document cloud presence at ~12km ASL 60km away from the plane, at**

**~10km ASL 120km away, down to ~6km ASL 330km away, and then up again. Due to refraction, each line of sight is not a perfect straight line.**

The lines of sight should be straight lines, but due to the refraction in the atmosphere, they are slightly curved. When plotted in a reference system with earth surface as a straight line, the lines of sights adopt a parabolic form. The tangent point moves away from the observer with decreasing tangent point altitude and the longest line of sight is the lowest one. Please see the answer to the previous point of this comment.

**My understanding of those points sometimes comes from later sections of the text. If my understanding is correct, could you please find a way to include those points in the text before reaching Figure 2, or in the legend of Figure 2, to help readers unfamiliar with the approach?**

In order to provide a clearer description of the viewing geometry, we have included the points of this comment in the caption of Figure 2:

"a) Example of the measuring geometry of GLORIA. The tangent altitude point is the closest point of the LOS (line-of-sight) to the surface. b) Example of the measuring geometry of GLORIA in a Cartesian system. Each blue line represents a different LOS that corresponds to a different elevation angle. The LOS is not a perfect straight line due to the refraction in the atmosphere and when plotted in a reference system with the Earth's surface as a straight line, it adopts a parabolic form. Each LOS is associated to a different tangent altitude. The horizontal distance of each LOS can extend several hundreds of kilometers. The lowest LOS are the ones with the largest path. The radiance measured by GLORIA is integrated along each LOS and contains the information related to the presence of clouds. The radiance from all LOS are recorded simultaneously. The red line indicates the tropopause (TP) from ERA5 along the corresponding LOS, in dark blue."

Also Figure 2 has been modified, as we considered the following image can help to better visualize the geometry.

[Figure]

**14. Figure 2: Would it be possible to add a cloud to this figure and show next to it the cloud profile that would then be generated, with the final base and top altitudes? It would help interpretation by non-experts.**

The aircraft is at ~14 km. If we assume a 1D cloud layer horizontally infinite, which could be a representation of a large scale cirrus field, indicated by the thick blue line, the CTH would be referenced at 11 km.

[Figure]

**15. Figure 2: What happens when a cloud is present at high distances from the plane, beyond the tangent, and is picked up by a line-of-sight that shoots low? For instance, a cloud located 600km away from the plane at 14km ASL could appear in the lowest line of sight. In that case, would the cloud be attributed a ~6km altitude, ie a 8km error? Is there a way to know whether that situation is frequent, i.e. to quantify the error in vertical distribution due to that uncertainty on the cloud position along the line of sight?**

It is right that if a cloud is located at 14 km altitude and is picked up by the line of sight with a tangent point altitude of 6 km, the cloud will be attributed to 6 km. In principle, it is not possible to know how many times that situation has been encountered in our analysis of the WISE campaign data. There is however, a study that could serve as a reference for the uncertainty in the CTH derived from limb viewing instruments and to which the reader is referred in page 11, line 16. Kent et al. (1997) address the questions than can arise from using the limb technique and the assumptions done in the retrieval. To do so, they simulated SAGE II cloud measurements and studied the uncertainty in the CTH determination and possible error situations. As stated in this comment, the first error is that the true cloud might be located at a higher altitude than that of the tangent layer to which is attributed. In the study of Kent et al. (1997) about 60% of the simulations had no altitude error and under 40% showed an error of 1 km or greater.

**16. Section 2.1: This might be painfully obvious to the authors, but I'm afraid I'm not sure I understand in which direction the scan is occurring? Are the lines of sight always pointing the same way with respect to the plane? Is the instrument looking in a different direction every time the plane changes direction?**

GLORIA is always pointing to the horizon from the right side of the plane. There are three measuring modes. If the instrument is set in chemistry mode, then all lines of sights are 90°

(perpendicular to the flight trajectory). If the instrument is in premier or panorama or panorama mode, then it pans from 45° to 135° (with respect to the plane) in intervals of 4° and 2°, respectively.

[Figure]

The paragraph in page 4, line 4-9 has been modified to make this point clearer:
"GLORIA always points towards the horizon from the right side of the plane. It is typically configured to one of three measuring modes: one high spectral resolution mode called chemistry mode (CM) and two modes, premier and panorama modes (DM), focusing on dynamical effects in the atmosphere. During CM, the instrument is fixed at 90° with respect to the flight trajectory. During the premier and panorama mode, the instrument changes its viewing direction between 45° and 135° in steps of 4° and 2°, respectively, which gives the possibility of observing the same volume of air from different perspectives and thus allowing for tomographic studies."

**17. P. 9, Figure 3: Assuming I've understood correctly how to interpret GLORIA's measurements, does it mean that cloud base and top can be located very different distances away from the plane? Could you comment on whether this might impact the retrieved vertical distribution somehow?**
If the observer is at 14 km, and the cloud top is assigned to the tangent point at 12 km and the cloud bottom at 10 km, that would mean a distance from the observer of about 150 km and 240 km, respectively. In the case of a homogenous cloud layer, this would not affect the vertical distribution. A more complicated case, are patchy clouds, but both cloud top and cloud bottom would be still relatively correct.

**18. p. 9, L. 7: "the clouds were referred to the tangent point": Is this assumption made for convenience in absence of useful information, or is there a reason to think it is realistic? Is there a way to verify how frequently this assumption is reasonable or not, and quantify what the related uncertainty means for the cloud profiles?**
In the retrieval technique applied to limb instruments, the inverse problem, necessary to retrieve the extinction from the measured radiance, is solved by dividing the atmosphere in spherical layers that are assumed homogenous, i.e. that the cloud would fill the complete layer:

[Figure]

 As discussed in comment 15, this assumption would lead to question the validity of the interpretation of the results (please, see answer to comment 15).

The leading error source in the determination of the cloud top, is the pointing knowledge along the LOS, which is about a tenth of a degree (Page 9, line 10-12). Other source are the oscillations in the extinction profile caused by the Gibbs phenomenon The Gibbs phenomenon describes the behavior of a Fourier series at a jump discontinuity, where an overshoot occurs and it does not disappear even if more terms are added to the Fourier sum. The overshoot, i.e. the radiance value larger than the maximum of the step function is of $\approx 10$ %. At the edges of the function and after the overshoot, ringing artifacts, i.e small oscillations, appear. These effects can cause an error in the determination of the cloud top height of one grid point ($\pm 125$ m). These oscillations could also affect the determination of the cloud bottom, creating a false detection of a thin layer ($1 - 2$ pixels) above a thick cloud in $\approx 1$ % of all the cloudy profiles. (Page 9, line 6-10).

**19. Figure 3 and others: The WISE flights seem to cover a very large latitude range, from polar to almost tropical regions. Given we don't know where are clouds along the LOS, and what we know about the direction the LOS are pointing to, and the variable flight paths followed by the plane, is there a way to document the horizontal area described by these results?**

As a reference, the location of the tangent altitude points has been included in the overview of scientific flights of the WISE campaign in Figure 1. The caption of Fig. 1 has been modified accordingly: "Overview of the 15 scientific flights of the WISE campaign. Color points correspond to the positions of HALO with GLORIA measuring. The red star indicates Oberpfaffenhofen, Germany and the red triangle Shannon, Ireland. The shade in light yellow gives a reference of the area covered by the measurements, indicating the distance of the tangent altitude point, i.e. of the closest point of the line of sight of the instrument to the surface."

[Figure]

**20. P. 10, Figure 5: could you add a figure showing the distribution of CI vs. Extinction? It could help quantify whether both variables are interchangeable**.

Figure 5 has been modified to the following.

[Figure]

CI and extinction do not follow a one-one relation for the whole range, but for CI between 3 and 5, which would correspond to optically thin clouds (Spang et al., 2008), the relation is stronger (page 11, line 2-3).

Following the suggestion of referee 2, the percentage over all profiles that show a CTH for both methods is added in page 13, line 9-10. 61% of all profiles show a CTH for the extinction method, 59% for the CI and 58% for both methods. Therefore, they yield similar cloud identification results.

**21. P. 12, L 26: "as discussed in Pan and Munchak (2011)..." The definitions of the tropopause are important, but the geographic colocation of the cirrus and the retrieved tropopause altitude is important as well. Given it is not possible to tell where a detected cloud is along the line of sight, it means there is an uncertainty of several hundred kms on the horizontal location of the detected cloud, with the largest uncertainties at the highest altitudes (closest to the tropopause). Trying to compare the altitude of a cirrus with the altitude of the tropopause when both can be separated horizontally by 600km makes me nervous, especially for clouds which are close to the midlatitude tropics boundary (where the tropopause can vary by several kms in a few hundred kms). Could you comment on the impact the cloud location uncertainty can have on colocating it with the tropopause, taking into account the latitude?**

In order to take into account the variability of the tropopause along the line of sight, we decided to provide two altitudes for the tropopause: the median and the percentile 95 computed for each line of sight. The median would correspond to a more relax criteria, whereas the percentile 95 is a very conservative approach. Considering that the altitude of the determined cloud top could be lower than the true one (see answer to comment 15) if a cloud top is above the percentile 95, it gives us confidence enough to conclude that cloud tops were observed above the tropopause, despite the uncertainty in its exact location.

The following table is provided with the intention of giving a deeper insight into the variability of the tropopause for each flight, taking into account the maximum, the minimum, the median and the percentile 95. The maximum of the difference max – min (of all flights) is around 8 km, however, the median of the max-min is about 1 km for each flight, indicating that such a big difference is not often encountered.

|  | med(Max-min) | med(max-95) | med(95-med) | med(med-min) |
|---|---|---|---|---|
| **F02** | 0.6 | 0.02 | 0.18 | 0.23 |
| **F03** | 0.8 | 0.02 | 0.23 | 0.43 |
| **F04** | 0.59 | 0.02 | 0.11 | 0.33 |
| **F05** | 1.02 | 0.05 | 0.34 | 0.44 |
| **F06** | 0.77 | 0.02 | 0.2 | 0.46 |
| **F07** | 1.28 | 0.03 | 0.5 | 0.6 |
| **F08** | 0.79 | 0.03 | 0.3 | 0.41 |
| **F09** | 0.9 | 0.02 | 0.35 | 0.45 |
| **F10** | 0.9 | 0.02 | 0.37 | 0.28 |
| **F11** | 0.39 | 0.02 | 0.16 | 0.17 |
| **F12** | 1.38 | 0.03 | 0.6 | 0.4 |
| **F13** | 1.02 | 0.04 | 0.37 | 0.32 |
| **F14** | 2.1 | 0.05 | 0.5 | 0.98 |
| **F15** | 1.05 | 0.02 | 0.46 | 0.41 |
| **F16** | 0.37 | 0.01 | 0.13 | 0.16 |
| **Mean(All flights** | 0.93 | 0.03 | 0.32 | 0.4 |
| **Max of max all** | 8.3 | 2.94 | 7.52 | 6.8 |
| **Min of min all** | 0.03 | 0 | 0.008 | 0.01 |

**22. P. 13, Figure 7: Could you maybe plot the zonally averaged tropopause altitude on these plots?**
We have added the mean tropopause height in coordinates of equivalent latitude to each plot in Figure 7. The tropopause has been computed from the ECMWF analysis data set, averaged over the duration period of the WISE campaign.

The caption of Figure 7 has been modified accordingly:
"PDFs of CTH as function of equivalent latitude (Eqlat) normalized for each altitude bin from (a) the extinction, (b) the CI and (c) ERA5 (discussed in Sect 4.3). The y axis shows the altitude of the tangent points (TgPt). The black line represents the mean tropopause height during September-October 2017 as a function of the equivalent latitude. It was computed from ECMWF analysis data."

[Figure]

**23. P. 14, Figure 8: Would it be possible to separate the distribution of cloud properties depending on whether the clouds were observed in tropical, midlat or polar latitudes? I would expect the vertical extents to vary with latitude.**

The range of latitudes of the measurements covers from about 35°N to 75°N, which should not be mistaken with the equivalent latitude range than extends from 0°N - 90°N. The following plots correspond to the vertical extent of clouds in four latitude bands: [30-45)°N, [45-55)°N, [55-65) °N and [65-80] °N, for both detection methods. The latitude corresponds to the latitude of the tangent altitude point to which the CTHs are referred. N indicates the total number of CTHs, whereas n indicates the number of CTHs in the selected latitude band. The percentage is computed over N.

Most of the CTHs are located in the band 45-65°N and shifted towards vertical extension < 1.5 km.

We consider that these new plots look similar to the original Fig.8. Therefore, we have not modified Fig. 8 and we have added in page 15, line 6 that the latitude band with the largest number of CTH is between 45-65°N.

[Figure]

**24. P. 14, L.5-6: "Spang et al..." this has already been stated close to the introduction. Please avoid repetition.**
The sentence in question is the following: 'Spang et al. (2015) analyzed CRISTA data (Spang et al., 2015) and concluded with a frequency of occurrence of 5% of all observations and Zou et al. (2020) obtained 2% for CALIPSO data and 4 – 5% for MIPAS data.'
In the introduction the sentence is: 'A recent study with the Michelson Interferometer for Passive Atmospheric Sounding (MIPAS) and Cloud-Aerosol Lidar and Infrared Pathfinder Satellite Observations (CALIPSO) by Zou et al. (2020) finds that CALIPSO observes occurrence frequencies of about 2% of stratospheric cirrus clouds at mid and high latitudes and 4 – 5% for MIPAS in middle latitudes.'

We consider this repetition to be justified. In the introduction is used to give context to this study and further in the article to compare our results with their results, without needing to go back to the intro. We have included the results of these others studies in Table 2 to facilitate the comparison between results.
The sentence in question refers twice to Spang et al. (2105) and one of the references has been omitted (Page 16, line 26).

**25. P. 15, Figure 10: I'm not sure I understand what is to be learned from these figures. If I understand correctly, they show the spatial distribution of detected cloud tops. Are we supposed to compare the left figure that shows all CTH with the right figure that shows only the CTH above the tropopause, and infer the regions where CTH above tropopause occur most frequently? If that is the case: 1) why change the color code between both figures? It makes the comparison much harder. Is the color code showing another point of comparison, or another independent information? 2) Isn't there an easier way to represent the density of CTH above the tropopause relative to the total number of measurements? (Eg counting the number of cloud points in 1x1° or 2x2° boxes?) 3) why isn't that discussed in the text? I don't understand how Figure 10 relates to what is discussed P. 13 lines 8-12.**
The intention of figure 10 is to show the distribution of CTHs during the WISE campaign (a) and the distribution of the CTHs above the tropopause (b). Following the suggestion, we have changed the color code, so both figures have the same information (altitude of the CTHs). Figure 9 gives the occurrence frequency of CTHs above the median tropopause relative to the total number of observation.
We have modified the discussion about Fig. 10:
"Figure 10 shows the distribution of all CTHs (Fig.10) and the distribution of CTHs above $TP_{med}$ for the extinction method. As can be seen, most of the occurrences of CTHs above $TP_{med}$ were found between 50-70°N, with varying altitudes from 8-13 km. The few occurrences between 35-50°N were located at higher altitudes, from 10-14 km." (Page 16, lines 15-18)

**26. P.16, L. 12: The already stated CALIPSO references (Martins and Reverdy) show a larger frequency of SVC and focus on the Tropics, it might make sense to refer to them.**
We consider that introducing these references in the suggested section, i.e, comparison of GLORIA with ERA5, would affect the story line. We have added a reference to these studies in the introduction, where we can give an overview of the knowledge about SVC also in the tropics.

[revised manuscript text omitted]